# Extra-Merge: Tracing the Rank-1 Subspace of Model Merging in Language Model Pre-Training

Wenjie Zhou [1 2 *]   Bohan Wang [3 *]   Hongtao Zhang [1 4]   Chenxi Jia [1 5]   Wei Chen [1 2]   Xueqi Cheng [1 2]

## Abstract

Model merging has emerged as a lightweight paradigm for enhancing Large Language Models (LLMs), yet its underlying mechanisms remain poorly understood. In this work, we analyze late-stage pre-training trajectories and uncover a **Rank-1 Subspace** phenomenon: while raw optimization steps oscillate violently, consecutive *merged* checkpoints collapse onto a stable, approximately one-dimensional linear manifold. We theoretically ground this observation in a *river-valley* landscape analysis: averaging acts as a geometric low-pass filter that dampens high-curvature noise to reveal the optimal descent direction. Capitalizing on this insight, we propose **Extra-Merge**, a training-free strategy that extrapolates along this subspace to minimize loss without additional gradient updates. Extensive experiments across GPT-2 and LLaMA families (124M to 2B) demonstrate that Extra-Merge consistently outperforms standard merging baselines. Notably, it yields consistent zero-shot accuracy gains on Pythia-12B downstream tasks and generalizes effectively to the Muon optimizer (Jordan et al., 2024).

## 1. Introduction

Pre-training Large Language Models (LLMs) is characterized by rigid optimization schedules and immense computational costs (OpenAI et al., 2024; Comanici et al., 2025).

Modern pre-training runs consume vast GPU hours (Kaplan et al., 2020; Hoffmann et al., 2022; Hägele et al., 2024), with the optimization trajectory strictly dictated by learning rate schedulers—such as Warmup-Stable-Decay (WSD) (Hu et al., 2024) or Cosine annealing. Consequently, the final model capability is often path-dependent, realized only during the terminal phase of the learning rate decay. Given these constraints, there is a critical demand for *post-hoc* mechanisms that can exploit the latent information within the optimization trajectory to maximize performance, without incurring the prohibitive cost of additional gradient steps.

In response, model merging has emerged as a potent, training-free paradigm to address this efficiency bottleneck. Rooted in classical Polyak averaging (Polyak & Juditsky, 1992) and revitalized by Stochastic Weight Averaging (SWA) (Izmailov et al., 2018), this approach posits that averaging points along the optimization path approximates the center of a flat local minimum, thereby enhancing generalization. Recently, frameworks like Latest Weight Averaging (LAWA) (Kaddour, 2022) and Pre-trained Model Averaging (PMA) (Li et al., 2025) have successfully adapted this paradigm to LLM pre-training. By aggregating late-stage checkpoints via uniform or exponential moving averages, these methods consistently reduce validation loss and bolster performance on downstream reasoning tasks, offering a "free lunch" improvement over the final converged model.

Despite model merging's widespread adoption, the geometric principles underpinning model merging in the nonconvex landscape of LLMs remain obscure. While existing literature attributes these gains to the static property of *local flatness* (Izmailov et al., 2018; Chen et al., 2017; Li et al., 2025), this perspective overlooks the dynamic geometry of the optimization trajectory itself. It remains unclear how the merging process transforms the chaotic oscillation of raw gradients into a coherent path of descent. We aims to bridge this gap by shifting the focus from the static loss landscape to the *trajectory geometry*, addressing a fundamental question:

*What are the geometric properties of the manifold formed by merged checkpoints, and can we exploit this structure for futher enhancement?*

---

*Equal contribution  [1] State Key Laboratory of AI Safety, Institute of Computing Technology, Chinese Academy of Sciences, China [2] University of Chinese Academy of Sciences, China [3] Alibaba Group, China [4] School of Advanced Interdisciplinary Sciences, University of Chinese Academy of Sciences, Beijing, China [5] School of Mathematics, Southeast University, Nanjing, China. Correspondence to: Wei Chen <chenwei2022@ict.ac.cn>, Wenjie Zhou <zj4323005@gmail.com>, Bohan Wang <bhwangfy@gmail.com>, Xueqi Cheng <cxq@ict.ac.cn>.

*Proceedings of the 43rd International Conference on Machine Learning*, Seoul, South Korea. PMLR 306, 2026. Copyright 2026 by the author(s).

Our contributions are summarized as follows:

- **Rank-1 subspace phenomenon in merged trajectories.** We empirically show that late-stage *merged* checkpoints concentrate on an approximately one-dimensional linear manifold.Concretely, while interpolation between adjacent *raw* checkpoints exhibits a convex-basin profile (midpoint loss lower than both endpoints), interpolation between adjacent *merged* checkpoints becomes nearly monotone, indicating a smoother path of progress. Through extensive PCA analysis, we demonstrate that this **Rank-1 Subspace** captures the dominant direction of descent (explaining $> 94\%$ of variance), this contrasts sharply with raw trajectories,whose variance disperses over multiple components and whose projections are non-monotone.

- **Extra-Merge Strategy.** Capitalizing on this geometric insight, we propose Extra-Merge, a training-free algorithm that estimates the subspace tangent to extrapolate progress without additional gradient updates. We provide a rigorous theoretical justification under the *river-valley* loss framework (Wen et al., 2024), proving that averaging acts as a geometric low-pass filter that aligns the trajectory with the optimal descent direction, thereby guaranteeing loss reduction via extrapolation.

- **Universality across Scales and Optimizers.** Finally, We validate Extra-Merge across a wide range of model scales (GPT-2/LLaMA, 124M to 2B) and learning schedulers. Crucially, our method yields consistent zero-shot accuracy gains on Pythia-12B (Biderman et al., 2023) downstream benchmarks and generalizes effectively to the orthogonal update regime of the **Muon optimizer**, demonstrating that the Rank-1 Subspace is a robust, optimizer-agnostic property of LLM training.

**Conflict of Interest Disclosure.** The authors declare no financial conflicts of interest related to this work.

## 2. Related Works

**Model Merging in Pre-training.** Model merging is a classic topic in machine learning (Utans, 1996; Chen et al., 2017; Wortsman et al., 2022; Yang et al., 2024). In this paper, we specifically employ checkpoints saved during pre-training and average these checkpoint parameters to improve performance without requiring substantial resources. The modern resurgence of this paradigm in deep learning is largely attributed to **Stochastic Weight Averaging (SWA)** (Izmailov et al., 2018). SWA demonstrated that averaging weights traversed by SGD with a cyclical or high constant learning rate leads to wider optima and better generalization than standard training. More recently, these techniques have been adapted specifically for Large Language Models (LLMs). **Latest Weight Averaging (LAWA)** (Kaddour, 2022; Sanyal et al., 2023) and Pre-trained Model Averaging

(PMA) (Li et al., 2025) investigate the benefits of averaging late-stage checkpoints in pre-training. Similarly, recent studies (Tian et al., 2025; Liu et al., 2024; Luo et al., 2025) have integrated merging directly into the pre-training loop. Unlike prior works that view checkpoints as static points for interpolation, we interpret them as a dynamic sequence, allowing us to extrapolate beyond the convex hull of observed weights. To ensure a comprehensive evaluation of novelty and distinctiveness, we provide a detailed taxonomic analysis of the model merging landscape in Appendix A.

**Geometry of the Loss Landscape.** The geometry of the loss landscape is fundamental to understanding the success of deep neural networks (Fort & Ganguli, 2019; Gur-Ari et al., 2018; Li et al., 2018). Closely related to merging is the phenomenon of **Linear Mode Connectivity (LMC)** (Frankle et al., 2020; Gotmare et al., 2018; Goodfellow et al., 2014; Vlaar & Frankle, 2022; Lucas et al., 2021), which establishes that multiple models trained independently on the same task can often be connected via a low-loss linear path. While LMC typically addresses the connectivity between converged and independent models, our work investigates the local geometry of the trajectory *during* the late stages of a single pre-training run. Most relevant to our proposed Extra-Merge is the **River-Valley landscape** perspective on optimization dynamics (Cohen et al., 2021; Wen et al., 2024; Song et al., 2024; Davis et al., 2025). These works characterize the neural network loss landscape as being composed of flat directions ("rivers") and sharp directions ("mountains"), corresponding to Hessian eigendirections with distinct curvatures. Furthermore, recent studies have leveraged this intuition to propose effective acceleration strategies for LLM pre-training (Wang et al., 2025; Zhou et al., 2025; Wang et al., 2024), providing empirical validation for the correctness of this perspective.

## 3. Preliminaries

**Problem Setup.** We consider the pre-training of a Large Language Model (LLM) parameterized by $\theta \in \mathbb{R}^d$. The optimization process generates a trajectory of parameters $\{\theta_0, \theta_1, \ldots, \theta_T\}$ by minimizing a loss function $\mathcal{L}(\theta)$ over a dataset, typically using optimizers such as AdamW. We denote the checkpoint saved at training step $t$ as $\theta_t$.

**The LAWA Framework.** Current research on model merging in pre-training is mainly captured by the *Latest Weight Averaging* (LAWA) framework (Kaddour, 2022; Sanyal et al., 2023). LAWA aggregates a sequence of $n$ checkpoints sampled with a fixed interval $\tau$, computing a weighted average:

$$\theta_{\text{avg}}^{(t)} = \sum_{i=0}^{n-1} w_i \cdot \theta_{t-i\tau}, \quad \text{s.t.} \quad \sum_{i=0}^{n-1} w_i = 1, \qquad (1)$$

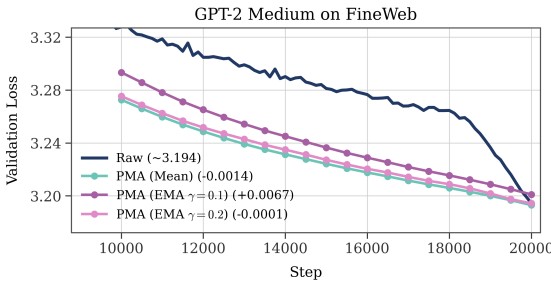

*Figure 1.* **Comparison of merging strategies.** We compare the validation loss of the raw optimization trajectory against PMA (Uniform) and EMA-based merging schemes ($\gamma \in \{0.1, 0.2\}$). Experiments are conducted on GPT-2 Small (124M) trained on FineWeb ($\tau = 500, n = 8$).

where $w_{i=0}^{n-1}$ are scalar weights determining the contribution of each checkpoint. This formulation allows for various weighting schemes, such as Exponential Moving Average (EMA), where $\theta_{\text{avg}}^{(t)} = \gamma \theta_t + (1 - \gamma)\theta_{\text{avg}}^{(t-\tau)}$.

**Baseline: Pre-trained Model Averaging (PMA).** While EMA is standard in convex optimization, recent work on *Pre-trained Model Averaging* (PMA) (Li et al., 2025) provides compelling evidence that simple **Uniform Averaging** (i.e., $w_i = 1/n$) is superior for LLM pre-training. PMA posits that in the high-dimensional, non-convex landscape of LLMs, uniform weights effectively approximate the centroid of the local loss basin, offering better stability than EMA.

To rigorously establish our baseline, we validate this observation in Figure 1. We compare the raw trajectory against LAWA instantiated with both Uniform weights (PMA) and EMA weights ($\gamma \in 0.1, 0.2$) on GPT-2 Small. Consistent with Li et al. (2025), Uniform Averaging yields the lowest and most stable validation loss. Consequently, we adopt PMA (Uniform Averaging) as the standard baseline throughout this work, denoted as $\theta_{\text{avg}}^{(t)} = \frac{1}{n} \sum_{i=0}^{n-1} \theta_{t-i\tau}$.

# 4. Main Findings

## 4.1. Warmup: Pairwise Connectivity

To unravel the geometric mechanism of model merging, we begin by probing the connectivity between *adjacent* checkpoints. This serves as a fundamental test: if the optimization trajectory is smooth, linear interpolation between steps should yield monotonic improvement. Conversely, non-monotonicity implies curvature or oscillation.

We analyze the validation loss $\mathcal{L}(\theta(\alpha))$ along the linear path $\theta(\alpha) = (1-\alpha)\theta_t + \alpha\theta_{t+\tau}$ for $\alpha \in [0, 1]$. Figure 2 contrasts the landscape of the raw trajectory ($\theta^{\text{raw}}t \rightarrow \theta^{\text{raw}}t + \tau$) against the merged trajectory ($\theta^{\text{avg}}t \rightarrow \theta^{\text{avg}}t + \tau$) on GPT-2 Small under the same setting of Figure 1. Two distinct geometric regimes emerge:

- **Raw checkpoints exhibit Convex Basins.** As shown in Figure 2 (Left), the interpolation between raw checkpoints consistently forms a U-shaped convex basin. Notably, the midpoint loss ($\alpha \approx 0.5$) is significantly lower than both endpoints. This phenomenon suggests that consecutive raw steps are not moving strictly "downhill" along the valley floor. Instead, they oscillate across the high-curvature walls of the loss valley, effectively "bouncing" around the optimal path.

- **Merged checkpoints reveal Monotonic Descent.** In sharp contrast, Figure 2 (Right) reveals that the path between merged checkpoints is strictly monotonic. The loss decreases smoothly from $t$ to $t + \tau$ without the barrier or basin structure observed in the raw trajectory. This suggests that the averaging operation acts as a geometric filter, effectively canceling out the orthogonal oscillations.

**Implication.** The shift from local convexity to monotonicity indicates that merged checkpoints do not merely reside in a lower-loss region; they lie on a geometrically distinct, smoother manifold. This implies that while the raw trajectory is chaotic, the *merged* trajectory aligns with the monotonic direction, hinting at the existence of a stable, linear structure. In the following section, we extend this pairwise analysis to multiple checkpoints to uncover the vaster geometry of this trajectory

## 4.2. Rank-1 Subspace of Merged Models

To rigorously verify whether the local smoothness observed in Section 4.1 translates to a global low-dimensional manifold, we analyze the spectral geometry of trajectory segments. Specifically, we collect sequences of $K$ consecutive checkpoints, form the centered parameter matrix $\Theta \in \mathbb{R}^{d \times K}$, and perform Principal Component Analysis (PCA). We quantify the geometric structure using two metrics: the Explained Variance Ratio (EVR) of the $k$-th principal component ($R_k = \sigma_k^2 / \sum_j \sigma_j^2$) to measure linearity, and the Projected Coordinates along principal axes to assess monotonicity. We conduct this analysis across GPT-2 (Small/Medium) and LLaMA (0.5B/2B) families, comparing raw trajectories $\theta_t^{\text{raw}}$ against merged ones $\theta_t^{\text{avg}}$ ($\tau = 500, n = 8$) with a default segment length of $K = 5$.

**Phenomenon 1: Spectral Concentration.** As shown in Figure 3 (Top), merged checkpoints exhibit a dominant Rank-1 structure, consistently concentrating $> 94\%$ of variance in the first principal component ($R_1$) across all model scales. In contrast, raw trajectories display a dispersed spectrum with significant energy distributed across other components. This phenomenon indicates that averaging effectively sup-

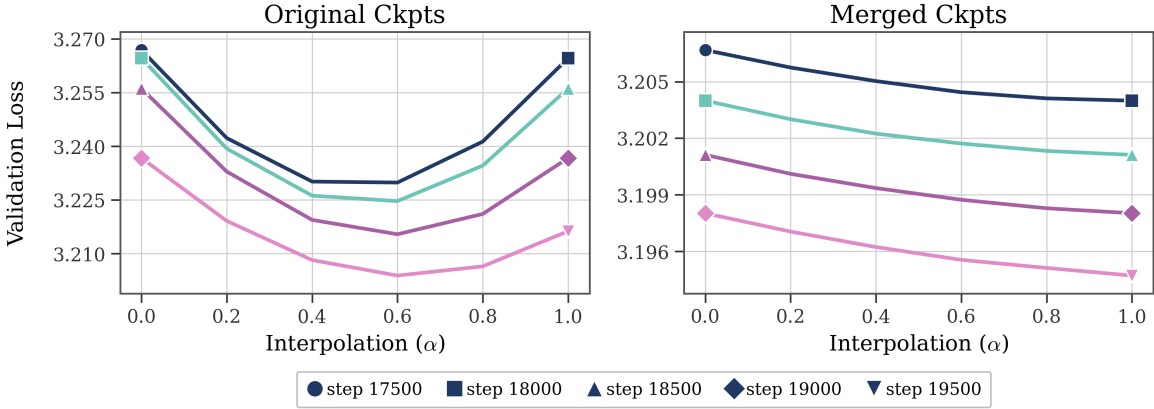

*Figure 2.* **Geometric shift from Convexity to Monotonicity.** We visualize the loss landscape along the linear interpolation path $\theta(\alpha) = (1 - \alpha)\theta_t + \alpha\theta_{t+\tau}$ between consecutive checkpoints ($\tau = 500$). **Left (Raw):** The trajectory exhibits a *convex basin* profile, where the midpoint loss is significantly lower than the endpoints. **Right (Merged):** The landscape transforms into a *monotonic descent*, under the merging strategy of ($\tau = 500, n = 8$).

presses orthogonal variance, isolating a single dominant axis of variation.

**Phenomenon 2: Trajectory Rectification.** Figure 3 (Bottom) visualizes the evolution of normalized coordinates projected onto the first principal direction $u_1$. While raw checkpoints exhibit non-monotonic oscillation, merged checkpoints evolve in a strictly monotonic, quasi-linear fashion. which implies that the extracted principal direction is not an artifact of noise but aligns with a coherent, directed path of descent.

**Phenomenon 3: Persistence over Horizons.** We further assess the stability of this structure by extending the trajectory length $K$. Figure 4 reveals that while the linearity of raw trajectories ($n = 1$) decays rapidly with $K$, merged trajectories with sufficient window sizes ($n > 4$) maintain robust linearity ($R_1 > 0.8$) even over long horizons. This suggests that with sufficient smoothing, the Rank-1 Subspace acts as a stable, global backbone of the late-stage training dynamics.

Above findings highlight a fundamental geometric shift: model merging effectively suppresses the high-frequency oscillations of the training trajectory, collapsing the optimization path onto a **Rank-1 Subspace**: a quasi-linear 1D manifold that aligns with the primary direction of loss descent.

**Is Linearity Trivial under Smoothing?** One might hypothesize that the observed Rank-1 structure is a trivial artifact of the sliding window, which introduces temporal autocorrelation via data overlap. However, smoothing does not inherently induce rank collapse. Consider a null hypothesis where the trajectory is an isotropic random walk in $\mathbb{R}^d$: sliding window averaging acts as a scalar contraction on the covariance eigenvalues but preserves the effective rank (i.e.,

the trajectory "cloud" shrinks but remains spherical). In our experiment Figure 3, the emergence of a dominant principal component ($R_1 > 94\%$) implies a latent linear drift. We provide a formal justification for this phenomenon under the **River-Valley** loss landscape framework in Section 5.

## 5. Extra-Merge

Leveraging the geometric insight from Section 4.1 where merged checkpoints concentrate on a linear manifold, we propose **Extra-Merge**, a training-free algorithm designed to exploit this geometric stability. Unlike standard merging which interpolates *within* the convex hull of past checkpoints, Extra-Merge extrapolates the optimization trajectory along the estimated *Rank-1 Subspace* to minimize loss. Given a sequence of merged checkpoints $\theta_i^{\mathrm{avg}}$ generated by a fixed PMA schedule (interval $\tau$, window $n$), the procedure operates in two phases:

**Step 1. Extrapolation Direction Estimation.** We apply PCA to a sliding window of the $K$ most recent merged checkpoints, denoted as $\Theta_K = [\theta^{\mathrm{avg}}t - K + 1, \ldots, \theta^{\mathrm{avg}}t]$. We extract the first principal component $u_1 \in \mathbb{R}^d$ and orient $u_1$ to align with the temporal evolution of training. We define the extrapolation direction $\hat{v} = \mathrm{sgn}(\langle\theta^{\mathrm{avg}}t - \theta^{\mathrm{avg}}t - 1, u_1\rangle) \cdot u_1$, ensuring the direction points towards descent.

**Step 2. Adaptive Greedy Line Search.** Starting from the latest merged checkpoint $\theta^{\mathrm{avg}}t$, we perform a line search along $\hat{v}$. To avoid manual tuning of the search scale, we adopt an *adaptive step size* strategy based on the trajectory's local velocity. Let $z_i = \theta^{\mathrm{avg}}i \cdot \hat{v}$ be the projection coordinate. We define the base stride $\delta$ as a fraction $\alpha$ of the coordinate displacement between the last two merged states: $\delta = \alpha \cdot |z_t - zt - 1|$, with $\alpha = 0.1$ as a robust default. The

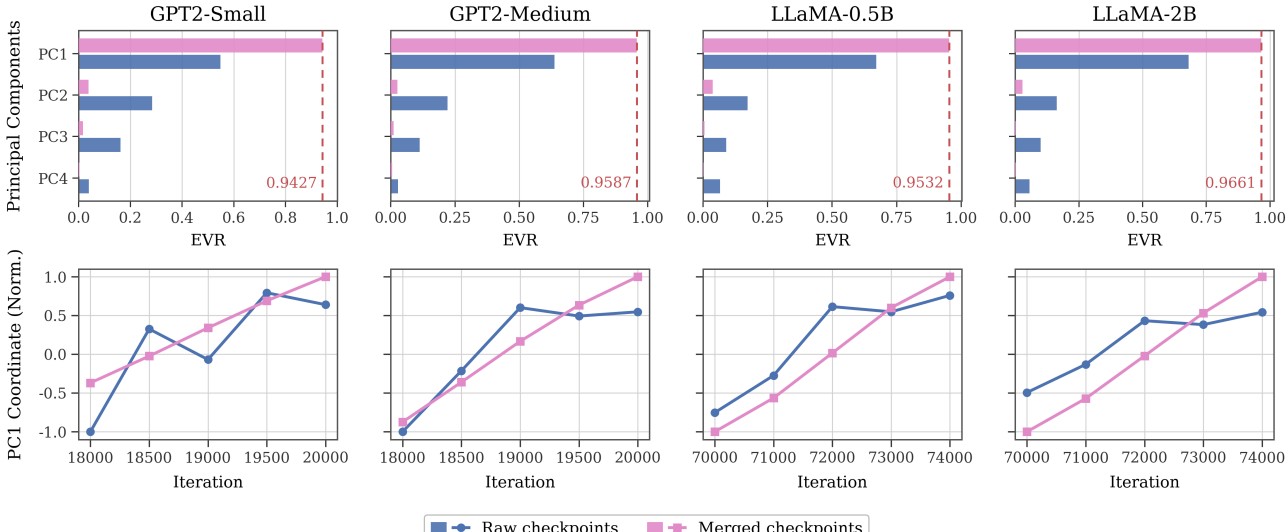

*Figure 3.* **PCA analysis of raw and merged trajectories.** We perform PCA on sequences of $K = 5$ consecutive checkpoints for GPT-2 and LLaMA models (From 124M–2B). **Top (Spectral Concentration):** Merged trajectories (pink) exhibit a dominant Rank-1 structure, with the first principal component (PC1) explaining $> 94\%$ of the variance. In contrast, raw trajectories (blue) disperse energy across multiple components. **Bottom (Trajectory Rectification):** Projecting checkpoints onto PC1 reveals that merging rectifies the optimization path into a strictly monotonic flow, whereas raw steps exhibit non-monotonic. This confirms that merging collapses the trajectory onto a stable 1D linear manifold.

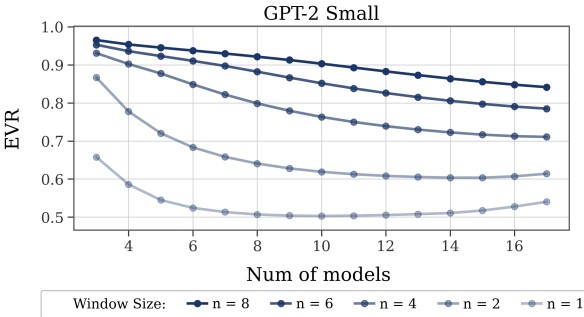

*Figure 4.* **Robustness of the Rank-1 Subspace.** We analyze the persistence of linearity (PC1 Explained Variance Ratio) as a function of the trajectory length $K$ (number of checkpoints included in PCA), varying the merging window size $n$ in GPT-2 Small experiment. Merged trajectories with sufficient window sizes ($n > 4$) maintain a robust linear structure ($EVR > 0.8$) even over long horizons in the late-stage of training ($K = 16$).

extrapolation is defined iteratively:

$$\theta_{\text{extra}}^{(k)} = \theta_t^{\text{avg}} + k \cdot \delta \cdot \hat{v}, \quad k = 1, 2, \dots \quad (2)$$

We evaluate the validation loss $\mathcal{L}(\theta_{\text{extra}}^{(k)})$ at each step and terminate the search immediately when the loss fails to improve relative to the anchor point (i.e., $\mathcal{L}(\theta_{\text{extra}}^{(k)}) > \mathcal{L}(\theta_t^{\text{avg}})$), returning the candidate with the minimal loss.

Figure 5 provides both a geometric interpretation and empirical verification of this procedure. For Figure 5a, while raw optimization steps oscillate across the sharp curvature of the loss landscape, merged checkpoints effectively collapse onto the valley floor, forming a stable Rank-1 Subspace. Extra-Merge exploits this stability to extend the trajectory

into the "undiscovered" region. Empirically (Figure 5b), we validate this behavior on GPT-2 Small ($\tau = 500, n = 8$) and estimating the subspace tangent from the last $K = 4$ merged checkpoints. The resulting loss curve exhibits a smooth continuation from the interpolation phase (green zone) to the extrapolation phase (gray zone), successfully locating a significantly lower-loss solution without additional gradient updates.

### 5.1. Theoretical Insight

In this section, we formalize the geometric intuition behind the **Rank-1 Subspace** and provide a theoretical justification for Extra-Merge. We demonstrate the underlying mechanism within the *River-Valley* framework (Wen et al., 2024), begin by defining the local geometry of late-stage pre-training. Throughout this section, $N$ corresponds to the merging window size $n$ in PMA/LAWA, and $T$ corresponds to the checkpoint interval $\tau$ used in practice. "River" refers to the (approximately) flattest direction, and "Mountains" refer to the remaining high-curvature subspace.

**Assumption 5.1** (River-Valley Landscape (Assumption 2 in (Wen et al., 2024)); Informal version of Assumptions C.1 and C.4). Consider a local neighborhood $U$ around the optimization path. We assume the loss $\mathcal{L}(\theta)$ decomposes into a 1D *flat* direction (the "river") and a $(d-1)$-dimensional *sharp* subspace (the "mountains"). Formally, let $v \in \mathbb{R}^d$ be the unit vector along the river, with associated projection matrices $P_F = vv^\top$ and $P_S = I - vv^\top$. For any $\theta \in U$,

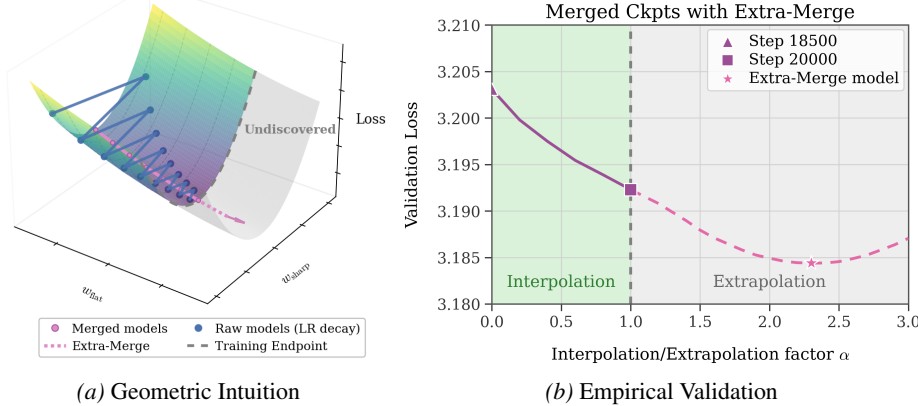

*(a)* Geometric Intuition       *(b)* Empirical Validation

*Figure 5.* **(a)** A visual illustration of the loss landscape. While raw optimization steps (blue trajectory) oscillate violently across the high-curvature valley walls, the merged checkpoints (pink nodes) filter out these oscillations, collapsing onto the stable valley floor (the Rank-1 Subspace). Extra-Merge (dashed arrow) identifies the tangent of this subspace to extrapolate towards the "undiscovered" low-loss region. **(b)** Validation loss profile on GPT-2 Small. The green zone represents interpolation between merged checkpoints, while the gray zone denotes the extrapolation phase. By extending along the principal direction with an adaptive step size, Extra-Merge locates a minimum (star) significantly lower than the final merged checkpoint.

the loss is approximated as:

$$\mathcal{L}(\theta) \approx \ell\big(v^\top(\theta - \theta_\star)\big) + \frac{1}{2}\|H^{1/2}P_S(\theta - \theta_\star)\|_2^2, \quad (3)$$

where $\theta_\star$ is a reference point on the river, $\ell(\cdot)$ is a slowly varying function along $v$, and $H$ is a positive semi-definite Hessian matrix supported on the mountain subspace (i.e., $H = P_S H P_S$) and satisfies $z^\top H z \in [\mu\|z\|_2^2, L_S\|z\|_2^2]$ for all $z \in \mathrm{range}(P_S)$.

Intuitively, Assumption 5.1 posits that the loss landscape resembles a narrow valley. The gradient aligns with the river direction $v$ only on the valley floor (the manifold $M$); elsewhere, it is dominated by steep components $HP_S(\theta - \theta_\star)$. Figure 5a is a 3-D illustration of this landscape. Next we model the late-stage behavior of Stochastic Gradient Descent (SGD) within this landscape.

**Assumption 5.2** (Decoupled late-stage dynamics; Informal version of Assumptions C.3, C.4). *Let $v \in \mathbb{R}^d$ be the (locally constant) river direction, Write the river and mountain coordinates as $t_k := v^\top(\theta_k - \theta_\star)$ and $z_k := P_S(\theta_k - \theta_\star)$. Assume the late-stage dynamics on a neighborhood $U$ can be approximated by*

$$t_{k+1} = t_k - \eta\big(\ell'(t_k) + \mu_F\big), \quad z_{k+1} = (I - \eta H)z_k + \eta g_k,$$

*where $\eta \in (0, 1/L_S)$, $H \succeq 0$ is supported on $\mathrm{range}(P_S)$ with eigenvalues in $[\mu, L_S]$, and $g_k$ are i.i.d. zero-mean noise supported on $\mathrm{range}(P_S)$ with $\mathrm{Cov}(g_k) = \sigma^2 P_S$ (e.g., $g_k \sim \mathcal{N}(0, \sigma^2 P_S)$).*

We now analyze the properties of merged checkpoints under these dynamics. Let $\theta^{\mathrm{avg}} = \frac{1}{N}\sum_{m=0}^{N-1}\theta_{t_0+mT}$ be the merged parameter obtained by averaging $N$ checkpoints sampled every $T$ steps. Our first result characterizes how

this averaging operation interacts with the landscape geometry.

**Theorem 5.3** (Averaging reduces expected distance to the river). *Let checkpoints be saved every $T$ steps and let $\theta^{avg}$ be the uniform average of $N$ late-stage checkpoints. Define the distance to the river as $D(\theta) := \|P_S(\theta - \theta_\star)\|_2$. Then*

$$\mathbb{E}\, D(\bar{\theta})^2 \;\leq\; \frac{1}{N}\Big(1 + \frac{2\varepsilon}{1-\varepsilon}\Big)\sum_{j=1}^{d-1}\frac{\eta\sigma^2}{2\lambda_j - \eta\lambda_j^2},$$

*where $\varepsilon := \max_j |1 - \eta\lambda_j|^T$, $\{\lambda_j\}_{j=1}^{d-1}$ are eigenvalues of $H$ on $\mathrm{range}(P_S)$. In particular, when $T$ is large enough such that $\varepsilon \ll 1$, the mountain deviation decays as $\tilde{\mathcal{O}}(1/N)$.*

Proof in Appendix C.2. Theorem 5.3 provides the justification for the "Rank-1 Subspace" phenomenon observed in Figure 3. Individual checkpoints $\theta_t$ are displaced from the valley floor by a constant noise floor proportional to $\eta\sigma^2/\lambda_{\min}$. In contrast, the averaging process suppresses this orthogonal oscillation by a factor of $1/N$. As $N$ increases, $\theta^{\mathrm{avg}}$ collapses onto the river manifold $M$.

Next, we address the validity of Extra-Merge. We analyze the sliding-window merging process, where we collect a sequence of $K$ merged checkpoints $\{\theta_s^{\mathrm{avg}}\}_{s=1}^K$. We perform PCA on these centered states to derive the extrapolation direction $\hat{u}_1$.

**Theorem 5.4** (PCA Recovers the Descent Direction). *Let $\hat{u}_1$ be the first principal component of a sequence of $K$ sliding-window merged checkpoints. Define the Signal-to-Noise Ratio (SNR) of the trajectory as $\rho := \sigma_{drift}^2/\sigma_{resid}^2$, where $\sigma_{drift}^2$ is the variance due to river descent and $\sigma_{resid}^2$ is the residual mountain noise after averaging. If $\rho > 1$, then*

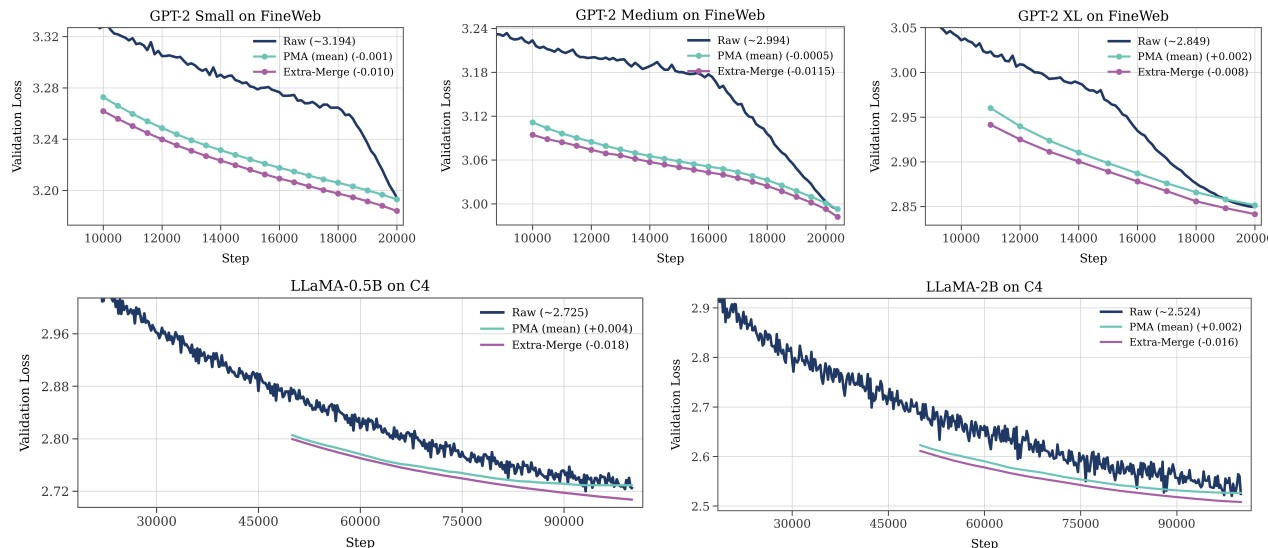

*Figure 6.* **Validation loss trajectories across model scales and schedules.** We compare Raw, PMA , and Extra-Merge on (First row) GPT-2 family trained with WSD scheduler, and (Last row) LLaMA family trained with Cosine scheduler. While PMA gradually aligns with the raw baseline as the learning rate decays, Extra-Merge achieve consistently lower loss.

*Table 1.* **Zero-shot performance on Pythia-12B downstream tasks.** We compare the raw checkpoint (at step 140k) against various merging strategies using checkpoints from 130k to 140k. Extra-Merge consistently outperforms both the raw baseline and standard averaging methods (PMA/EMA) across all four benchmarks.

| Task | Pythia-12B (Raw) | PMA (EMA$_{\alpha=0.1}$) | PMA (EMA$_{\alpha=0.2}$) | PMA (average) | Extra-Merge |
|---|---|---|---|---|---|
| Arc_challenge | 31.66 | 32.00 | 31.91 | 32.21 | **32.76** (+1.10) |
| Arc_easy | 69.74 | 69.11 | 69.78 | 70.08 | **70.58** (+0.84) |
| Hellaswag | 49.98 | 49.79 | 50.01 | 50.10 | **50.14** (+0.16) |
| PIQA | 76.12 | 76.12 | 76.06 | 75.90 | **76.38** (+0.26) |
| Average | 56.88 | 56.76 | 56.94 | 57.07 | **57.47** (+0.59) |

$\hat{u}_1$ *aligns with the true river direction* $v$:

$$\sin(\angle(\hat{u}_1, v)) \leq \frac{C}{\rho - 1}, \qquad (4)$$

*where* $C$ *is a constant determined by the residual noise level and standard PCA estimation stability.*

Proof in Appendix C.3. Theorem 5.4 is pivotal for Extra-Merge. It ensures that the PCA direction extracted from merged checkpoints is not a noise artifact: it recovers the tangent of the low-curvature subspace (the river direction $v$) once the drift-to-noise ratio $\rho$ is above 1. Therefore, performing a 1D line search along $\hat{u}_1$ acts as a training-free descent step on the rectified (merged) trajectory.

**Theoretical Implications for Hyperparameters.** To make the condition $\rho > 1$ concrete, we derive explicit bounds for the signal and noise terms in Lemmas C.9 and C.8 (due to space limit, we defer the theoretical result in Appendix C). In particular, the residual variance decreases roughly as $1/N$, while the drift signal grows with the temporal span $K\tau$. As a result, the effective SNR scales (up to con-

stants and mild correlation effects in $\tau$) as $\rho \propto N \cdot (K\tau)^2$. This gives the following guidance:

- **Window Size $N$ (Noise Suppression):** larger $N$ lowers the residual mountain noise floor (approximately $1/N$), making PCA less likely to lock onto noise, increasing $N$ can be helpful once the model is in a late training stage.
- **PCA Window $K$ (Signal Amplification):** increasing $K$ amplifies the cumulative drift along the river, improving identifiability of the subspace direction.
- **Interval $\tau$ (Span & Decorrelation):** a larger $\tau$ reduces temporal correlation in the mountain dynamics, improving the effectiveness of averaging (Theorem 5.3).
- **Trade-off:** $K$ (and $K\tau$) should remain moderate so the local "straight-river" approximation stays valid; then we use $K = 4$ in Figure 5b for balance.

## 6. Experiments

### 6.1. Experiment Settings

**Models and Datasets.** We conduct pre-training experiments on two widely adopted decoder-only families:

- **GPT-2 on FineWeb.** We train GPT-2 Small (124M), Medium (355M), and XL (1.55B) on the FineWeb dataset. This setup follows the highly optimized training recipe used in the nano-gpt benchmark[1], representing a rigorous standard for small-to-medium scale pre-training.

- **LLaMA on C4.** To verify scalability, we train LLaMA architectures with 0.5B and 2B parameters on the Colossal Clean Crawled Corpus (C4) dataset.

**Training Protocol.** All models are trained using the AdamW optimizer with hyperparameters $\beta_1 = 0.9, \beta_2 = 0.95$, and weight decay $\lambda = 0.1$. To ensure our method is agnostic to the learning rate (LR) schedule, we employ two distinct strategies: the WSD (Warmup-Stable-Decay) scheduler for GPT-2, and Cosine decay for LLaMA. This diversity confirms that the observed improvements are intrinsic to the geometry of merging. Detailed hyperparameters for all runs are provided in Appendix B.3.

**Merging Baselines and Extra-Merge Configuration.** We adopt average *Pre-trained Model Averaging* (PMA) as our primary baseline. To ensure a competitive comparison, we tune the merging interval $\tau$ and window size $n$, setting $\tau = 500, n = 8$ for GPT-2 Small/Medium, and $\tau = 1000, n = 10$ for the larger GPT-2 XL and LLaMA models, details of the merge settings can be found in Appendix B. We report results on the latter half of the training trajectory, where merging becomes effective. For Extra-Merge, we apply PCA over a sliding window of $K = 4$ consecutive checkpoints.

### 6.2. Main Results: Pre-training Validation Loss

We report the validation loss trajectories for both GPT-2 and LLaMA families in Figure 6, covering model scales from 124M to 2B. As illustrated, while PMA yields lower loss than the raw model during the high-learning-rate phase, its performance *gradually aligns with* the raw baseline as the learning rate decays. In contrast, **Extra-Merge** achieves a sustained loss reduction across all settings. By extrapolating along the optimization trajectory, our method consistently reaches lower loss values at equivalent training steps, effectively accelerating convergence without additional gradient updates. This represents a *free-lunch* gain, improving upon the raw baseline compared to PMA.

### 6.3. Scaling to Downstream Tasks: Pythia-12B

To assess the scalability and generalization capability of our method on larger-scale models and real-world tasks, we extend our evaluation to the **Pythia-12B** model. We

utilize the official intermediate checkpoints[2] ranging from step 130k to 140k, with a saving interval of 1k steps.

We compare Extra-Merge against the Raw baseline and PMA variants (Uniform and EMA) on four standard reasoning benchmarks: ARC-Challenge, ARC-Easy, HellaSwag, and PIQA. For Extra-Merge, we apply extrapolation along the principal direction extracted from the $K = 4$ merged checkpoints($\tau = 1000, N = 10$), and using average score as the indicator. We report the accuracy in Table 1.

**Results.** As detailed in Table 1, we observe that among standard merging strategies, PMA with uniform averaging generally yields the most robust improvements, outperforming EMA variants. Extra-Merge achieves an average accuracy gain of **+0.59%** over the raw baseline and **+0.40%** over the best PMA setting, confirming its efficacy on downstream tasks beyond simple perplexity reduction.

### 6.4. Extra-Merge on Muon Optimizer

To demonstrate the universality of Extra-Merge across different optimizers, we extend our evaluation to the **Muon** optimizer (Jordan et al., 2024). Unlike AdamW, Muon is characterized by its use of orthogonal updates, which induces a fundamentally different trajectory structure during training and are now widely adopted.

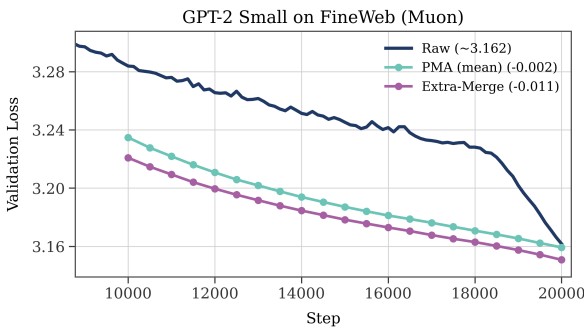

*Figure 7.* **Validation loss trajectory on FineWeb using the Muon optimizer.** We compare the Raw baseline, PMA, and Extra-Merge on GPT-2 Small. Extra-Merge consistently outperforms PMA.

We follow the experiment setting of the standard `modded nano-gpt` configuration. We maintain the identical merging protocol used in the AdamW experiments to ensure consistency. Detailed hyperparameters and experimental settings are provided in Appendix B.4. As illustrated in Figure 7, our method remains highly effective under Muon optimization. This confirms that the acceleration benefits of our method are robust to the underlying optimizer.

[1] https://github.com/KellerJordan/ modded-nanogpt

[2] https://huggingface.co/EleutherAI/ pythia-12b

# 7. Conclusion and Future Works

In this work, we identified the **Rank-1 Subspace**, a phenomenon where chaotic training trajectories collapse onto a stable 1D manifold upon averaging. Leveraging this, we proposed **Extra-Merge**, a training-free extrapolation strategy that consistently improves baseline merging performance across diverse architectures (GPT-2, LLaMA, Pythia) and optimizers (AdamW, Muon).

This work offers a novel geometric perspective on the linearity of LLM loss landscape. Future research could explore two key directions: (1) developing subspace-aware optimization strategies that actively align updates with this latent manifold to accelerate convergence, and (2) analyzing how this subspace reorients under different data distributions, which could unlock geometric approaches for efficient domain adaptation and transfer learning.

# Impact Statement

This paper contributes to advancing the field of deep learning, with a specific focus on understanding the training dynamics of Large Language Models (LLMs) and improving optimization efficiency through model merging techniques. Our research utilized a computational setup consisting of 8 NVIDIA RTX 4090 GPUs and 2 NVIDIA H100 GPUs, involving approximately 100 hours of training and inference time. By detailing these experiments, we hope to provide the community with new insights into the geometry of loss landscapes and the efficacy of merging strategies. As this work focuses on fundamental optimization methodologies, we do not foresee immediate negative societal impacts; rather, we aim to facilitate the development of more efficient and accessible AI systems.

# Acknowledgements

This work was funded by the Strategic Priority Research Program of the Chinese Academy of Sciences (Grant No. XDB0680101), CAS Project for Young Scientists in Basic Research under Grant No. YSBR-034, the National Key Research and Development Program of China under Grants No. 2023YFA1011602, and Xiaomi Young Talents Program.

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

## A. A Comprehensive Review of Model Merging Methods

**Post-Fine-Tuning Merging**. This topic focuses on integrating models fine-tuned on diverse tasks into a unified entity without incurring additional inference costs. Early heuristic approaches established the baseline: Model Soups demonstrates that averaging weights from hyperparameter sweeps enhances robustness and accuracy (Wortsman et al., 2022), while Fisher-Weighted Averaging refines this by weighting parameters based on their Fisher information (Matena & Raffel, 2022). Task Arithmetic interprets task-specific updates as vectors, enabling capability composition via linear operations (Ilharco et al., 2022). Addressing interference, TIES-Merging introduces trimming and sign-resolution steps to mitigate parameter conflicts (Yadav et al., 2023), and RegMean formulates merging as a linear regression problem with a closed-form solution (Jin et al., 2022). Moving towards adaptive methods, AdaMerging autonomously learns merging coefficients by minimizing the distance to task models on unlabelled test data (Yang et al., 2023b), whereas MetaGPT frames the objective as minimizing the average loss across tasks (Zhang et al., 2023). For automated discovery, Evolutionary Model Merge employs evolutionary algorithms to search for optimal merging recipes in large model spaces (Akiba et al., 2025). Finally, specialized techniques like DARE facilitate the merging of adapters by randomly dropping and rescaling parameters (Yu et al., 2024), and PAPA balances ensemble generalization with the efficiency of weight averaging (Ramé et al., 2023).

**Pre-training Model Merging**. In contrast to the above, this topic aims to accelerate convergence and improve generalization by exploiting the trajectory of intermediate model states. Fundamental to this approach, LAWA establishes that averaging recent checkpoints significantly reduces the training time required to reach target performance (Kaddour, 2022). Subsequent analysis in Early Weight Averaging reveals that models trained with high learning rates benefit disproportionately from averaging due to increased trajectory exploration (Sanyal et al., 2023). Empirical studies on Baichuan 2 validate these gains across different pre-training stages, highlighting the importance of checkpoint selection (Yang et al., 2023a). Theoretically, the WSM framework formalizes the relationship between learning rate schedules and model merging, proposing a "warmup-stable-merge" protocol (Tian et al., 2025). To optimize combination strategies, Dynamic Model Merging suggests weighting checkpoints based on generation quality metrics like perplexity, while other approaches utilize Bayesian Optimization to systematically search for the most effective weighting configurations (Liu et al., 2024). While large-scale model merging during pre-training is a nascent frontier, our comprehensive review identifies that simple averaging strategies exemplified by Pre-trained Model Averaging (PMA) constitute the optimal baseline practice. This approach is theoretically grounded in Polyak-Ruppert averaging (Polyak & Juditsky, 1992), thereby rendering our adoption of PMA as the primary baseline both sufficient and rigorous.

## B. Experiment Details

**Overview.** We evaluate Extra-Merge on (i) pre-training loss trajectories for GPT-2 on FineWeb and LLaMA on C4, and (ii) downstream accuracy for Pythia-12B using official intermediate checkpoints. Unless otherwise stated, we report the validation metrics of the *second half* of the training trajectory, where weight averaging becomes effective.

**Compute & implementation.** All experiments are run on 8×RTX4090-24G and 2×H100 with PyTorch 2.5.1, CUDA 12.2. Gradient clipping is set to 1.0 unless specified.

**Merging methods (common definitions).** Let $\theta_t$ be the raw checkpoint saved every $\tau$ optimizer steps. We define:

- **Raw:** use $\theta_t$ directly.

- **PMA (Uniform):** $\theta_t^{\text{PMA}} = \frac{1}{N} \sum_{i=0}^{N-1} \theta_{t-i\tau}$.

- **PMA (EMA):** $\theta_t^{\text{EMA}} = \sum_{i=0}^{N-1} w_i \theta_{t-i\tau}$, where $w_i \propto \gamma^i$ and $\sum_i w_i = 1$.

- **Extra-Merge:** (1) compute merged checkpoints $\theta_t^{\text{PMA}}$ as above; (2) run PCA on a sliding window of $K$ consecutive merged checkpoints to obtain $\hat{u}_1$; (3) extrapolate $\theta_t^{\text{EM}} = \theta_t^{\text{PMA}} + \alpha \hat{u}_1$.

We select the extrapolation coefficient $\alpha$ by a 1D line search] on the validation set using candidate values $\alpha \in \{[0,0.2,0.4,0.6,0.8,1,1.2,1.4,1.6,1.8,2.0]\}$, and report results using the best $\alpha$ once the validation loss (indicator) raise (unless stated otherwise).

## B.1. GPT-2 on FineWeb

**Models.** We train GPT-2 Small (124M), Medium (355M), and XL (1.55B) using the standard decoder-only Transformer architecture (Radford et al., 2019; Vaswani et al., 2017). To optimize computational efficiency, we follow the hyperparameter setting in NanoGPT speedrun benchmark.[3], which adopt specific settings for learning rates, weight decay, and scheduler phases for each model. The architectural details and corresponding hyperparameters are summarized in Table 2.

*Table 2.* GPT-2 architectures and hyperparameters used in our experiments. We decouple the learning rates for the language model head ($LR_{\text{head}}$) and the transformer body ($LR_{\text{body}}$). $T_{\text{warmup}}$ and $T_{\text{decay}}$ denote the number of steps for the warmup and decay phases of the WSD scheduler, respectively.

| Model | Params | $n_{\text{layer}}$ | $d_{\text{model}}$ | $n_{\text{head}}$ | $d_{\text{ff}}$ | $LR_{\text{head}}$ | $LR_{\text{body}}$ | WD | $T_{\text{warmup}}$ | $T_{\text{decay}}$ |
|---|---|---|---|---|---|---|---|---|---|---|
| GPT-2 Small | 124M | 12 | 768 | 12 | 3072 | $3.6 \times 10^{-3}$ | $1.8 \times 10^{-3}$ | 0.0 | 250 | 2000 |
| GPT-2 Medium | 355M | 24 | 1024 | 16 | 4096 | $3.0 \times 10^{-3}$ | $1.5 \times 10^{-3}$ | 0.125 | 500 | 4000 |
| GPT-2 XL | 1.55B | 48 | 1600 | 25 | 6400 | $7.5 \times 10^{-4}$ | $3.75 \times 10^{-4}$ | 0.0 | 500 | 6000 |

**Dataset.** We pre-train on the FineWeb dataset (Penedo et al., 2024), specifically utilizing the 10B token subset consistent with the NanoGPT speedrun benchmark. Tokenization utilizes the GPT-2 BPE tokenizer with a sequence length of $L = 1024$. We report validation loss on the standard FineWeb validation split (10M tokens).

**Training Protocol.** We train all models for a total of 20,000 steps. The global batch size is set to 512 sequences (approx. 0.52 million tokens per step), resulting in a total training budget of approximately 10 billion tokens. We utilize mixed-precision training with `bfloat16`.

**Optimization.** We use the AdamW optimizer (Loshchilov & Hutter, 2017) with $\beta_1 = 0.9$, $\beta_2 = 0.95$, and $\epsilon = 10^{-8}$. We employ a split optimization strategy where the embeddings and language model head are trained with a higher learning rate ($LR_{\text{head}}$) compared to the transformer backbone ($LR_{\text{body}}$), specifically setting $LR_{\text{body}} = 0.5 \times LR_{\text{head}}$. The learning rate follows a Warmup–Stable–Decay (WSD) schedule:

- **Warmup:** Linear warmup for $T_{\text{warmup}}$ steps (see Table 2).
- **Stable:** Constant learning rate at the peak value.
- **Decay:** Linear decay for the final $T_{\text{decay}}$ steps, reaching a minimum learning rate of $lr_{\min} = 0.05 \times lr_{\max}$.

Weight decay is applied only to the transformer body parameters for GPT-2 Medium ($\lambda = 0.125$), while it is set to 0 for Small and XL configurations based on empirical performance.

**Merge setting.** We save checkpoints every $\tau$ steps and apply merging on the latter half of training:

- GPT-2 Small/Medium: $\tau = 500$, $N = 8$.
- GPT-2 XL: $\tau = 1000$, $N = 10$.
- Extra-Merge: PCA window $K = 4$ consecutive merged checkpoints.

## B.2. LLaMA on C4

**Models.** We train LLaMA (Touvron et al., 2023) incorporating Rotary Positional Embeddings (RoPE) (Su et al., 2023), RMSNorm, and SwiGLU activations. We evaluate two specific scales: 0.5B and 2B. Using the code from HuggingFace Transformer Library (Wolf et al., 2020). The architectural configurations are detailed in Table 3.

**Dataset.** We pre-train on the C4 (Colossal Clean Crawled Corpus) (Raffel et al., 2020) dataset. Tokenization utilizes a SentencePiece tokenizer with a vocabulary size of $V = 32,100$. The sequence length is set to $L = 2048$. We follow the standard C4 validation protocol, reporting validation loss and perplexity on the official validation split.

---

[3] https://github.com/KellerJordan/modded-nanogpt

*Table 3.* LLaMA architectures and hyperparameters used in our experiments. $LR_{\max}$ denotes the peak learning rate used in the cosine schedule.

| Model | Params | $n_{\text{layer}}$ | $d_{\text{model}}$ | $n_{\text{head}}$ | $d_{\text{ff}}$ | $LR_{\max}$ |
|---|---|---|---|---|---|---|
| LLaMA-0.5B | 0.5B | 15 | 1280 | 20 | 5120 | $6.0 \times 10^{-4}$ |
| LLaMA-2B | 2B | 30 | 2048 | 32 | 8192 | $3.0 \times 10^{-4}$ |

**Training Protocol.** We follow the training setup of Wang et al. (2025); Zhao et al. (2024), utilizing the AdamW optimizer with $\beta_1 = 0.9$, $\beta_2 = 0.95$, weight decay $\lambda = 0.1$, and $\epsilon = 10^{-8}$. Gradient clipping is set to 1.0. The learning rate follows a cosine decay schedule:

- **Warmup:** Linear warmup for 1,000 steps.

- **Peak LR:** See Table 3 for model-specific values.

- **Decay:** Cosine decay to a terminal learning rate of $lr_{\min} = 0.1 \times lr_{\max}$.

The global batch size is set to 512 sequences. We train for a total of 100,000 steps, amounting to approximately 105 billion tokens.

**Implementation Details.** Models are trained using PyTorch Fully Sharded Data Parallel (FSDP). We employ a `HYBRID_SHARD` strategy with `bfloat16` mixed precision, where parameters and buffers are maintained in FP32 while gradients and communication reductions occur in `bfloat16`. We do not utilize CPU offloading.

**Merge setting.** For LLaMA models we use $\tau = 1000$, $N = 10$, and Extra-Merge uses $K = 4$ merged checkpoints.

### B.3. Pythia-12B Downstream Evaluation

**Model and checkpoints.** We evaluate **EleutherAI/Pythia-12B** (GPT-NeoX style; 36 layers, model dimension 5120, 40 heads) (Biderman et al., 2023) using the *official intermediate checkpoints* hosted on Hugging Face.[4] We use checkpoints from step 130k to 140k (inclusive) with a 1k-step interval, i.e., 11 checkpoints in total.

**Merging protocol on checkpoints.** We treat each 1k training step as one unit for checkpoint indexing and apply merging with:
$$\tau = 1000 \text{ (steps)}, \quad N = 10, \quad K = 4.$$
We compare Raw, PMA (Uniform/EMA), and Extra-Merge. For each method, we evaluate downstream tasks on the resulting merged/extrapolated checkpoint.

**Benchmarks and metric.** We evaluate on ARC-Challenge (Clark et al., 2018), ARC-Easy (Clark et al., 2018), HellaSwag (Zellers et al., 2019), and PIQA (Bisk et al., 2020), reporting accuracy (%). We compute the **average score** across tasks as the primary indicator. Evaluation is implemented with the LM Evaluation Harness (Gao et al., 2023), using:

- Few-shot setting: 0-shot

- Max sequence length: 2048

- Decoding: greedy (since these are multiple-choice log-likelihood tasks)

- Dtype: `float16`, batch size: 16

### B.4. Extra-Merge with Muon Optimizer

**Setting.** We additionally evaluate Extra-Merge under the **Muon** optimizer (Jordan et al., 2024; Liu et al., 2025), following the NanoGPT speedrun codebase configuration.[5] Muon differs from AdamW by employing orthogonalized updates via Newton-Schulz iterations, which substantially changes the geometry of the training trajectory.

---

[4]https://huggingface.co/EleutherAI/pythia-12b
[5]https://github.com/KellerJordan/modded-nanogpt

**Protocol.** We run GPT-2 Small on FineWeb with the same data pipeline and model configuration as in Section B.1, replacing AdamW with Muon. All merging hyperparameters are kept identical to the AdamW setting ($\tau = 500$, $N = 8$, $K = 4$) to isolate the effect of the optimizer. Other Muon-specific hyperparameters follow the default recipe in the benchmark codebase, including a split optimization strategy:

- **Transformer Body:** Optimized via Muon with learning rate $\eta = 0.02$ and momentum $\mu = 0.95$.

- **LM Head / Embeddings:** Optimized via AdamW with learning rate $\eta = 0.0036$.

### B.5. Large-scale PCA via Gram Matrix (Implementation Detail)

Directly computing PCA in parameter space is infeasible since the parameter dimension $d$ is extremely large while the number of checkpoints is small ($K \ll d$). We therefore compute the top principal component using the $K \times K$ Gram matrix.

Let $\{x_i\}_{i=1}^K$ be the centered checkpoints, where $x_i \in \mathbb{R}^d$ and $x_i = \theta_i - \frac{1}{K}\sum_{j=1}^K \theta_j$. Define the Gram matrix $G \in \mathbb{R}^{K \times K}$ as:

$$G_{ij} = x_i^\top x_j.$$

Let $q_1$ be the top eigenvector of $G$. The corresponding principal direction in parameter space is recovered by:

$$\hat{u}_1 = \frac{\sum_{i=1}^K (q_1)_i x_i}{\left\| \sum_{i=1}^K (q_1)_i x_i \right\|_2}.$$

This procedure is mathematically equivalent to PCA on the $d(\gg K)$-dimensional centered matrix but costs only $\mathcal{O}(K^2)$ inner products plus $\mathcal{O}(K^3)$ eigendecomposition, which is negligible for $K = 4$.

## C. Theoretical Insight: Model Merging as River Projection and Extra-Merge as Subspace Extrapolation

### C.1. Problem setting

Our setting follows the river-valley landscape formalism of Wen et al. (2024). We restate a self-contained version with a few additional assumptions tailored to checkpoint averaging and PCA.

**Notation.** Let $w \in \mathbb{R}^d$ denote parameters and $L : \mathbb{R}^d \to \mathbb{R}$ the loss. Let $\nabla L(w)$ and $\nabla^2 L(w)$ be the gradient and Hessian. Let $\lambda_1(H) \geq \cdots \geq \lambda_d(H)$ and $v_1(H), \ldots, v_d(H)$ denote eigenvalues/eigenvectors of a symmetric matrix $H$ in *descending* order; thus $v_d(\nabla^2 L(w))$ is the eigenvector of the *smallest* eigenvalue and is called the *flattest direction*. We write $v(w) := v_d(\nabla^2 L(w))$ and define the orthogonal projections $P_F(w) := v(w)v(w)^\top$ and $P_S(w) := I - P_F(w)$, where $F$ denotes the (one-dimensional) *river direction* and $S$ denotes the $(d-1)$-dimensional *mountain subspace*.

**River-valley geometry.** We assume a one-dimensional "river" manifold $M$ such that at points on $M$ the gradient aligns with the flattest direction.

**Assumption C.1** (River manifold and regularity, Assumption 2 in (Wen et al., 2024)). There exists a 1-dimensional $C^2$ manifold $M \subset \mathbb{R}^d$ and an open neighborhood $U \supset M$ such that:

1. **(Alignment on the river)** For all $w \in M$, $\nabla L(w)$ is parallel to $v(w)$.

2. **(Analyticity)** $L$ is analytic on $U$.

3. **(Bounded Hessian)** There exists $\beta_{\max} > 0$ such that $\|\nabla^2 L(w)\|_{\mathrm{op}} \leq \beta_{\max}$ for all $w \in U$.

4. **(Eigengap)** There exist $\beta > 0$ and $\beta_\flat > 0$ such that for all $w \in U$, $|\lambda_d(\nabla^2 L(w))| < \beta_\flat$ and $\lambda_{d-1}(\nabla^2 L(w)) > \beta + 4\beta_\flat$.

5. **(Slow spinning)** There exist $\rho \in [0, 0.01)$ and $0 < \nu_{\min} \leq \nu$ such that for all $w \in U$, $\nu_{\min} < \|\nabla L(w)\|_2^2 \leq \nu$ and $\|\nabla v(w)\|_{\mathrm{op}} \leq \rho \cdot \frac{\beta}{2\nu}$.

To obtain closed-form $(N, T)$-dependent bounds, we adopt the standard "straight river" simplification.

**Assumption C.2** (Straight river on $U$). On $U$, $v(w) \equiv v$ is constant, hence $P_F(w) \equiv P_F := vv^\top$ and $P_S(w) \equiv P_S := I - vv^\top$ are constant. The river is the affine line $M = \{w_\star + \alpha v : \alpha \in \mathbb{R}\}$ for some reference point $w_\star \in M$.

We model SGD with stochasticity only in mountain directions and a persistent, slow drift along the river. This drift accounts for the variance in the PCA signal.

**Assumption C.3** (SGD with drift and mountain-only noise). Let $\eta \in (0, 2/L_S)$ be a constant learning rate. The SGD iterates satisfy $w_{k+1} = w_k - \eta \nabla L(w_k) - \eta \mu_F v + \eta g_k$, where $(g_k)_{k \geq 0}$ are i.i.d. with $g_k \sim \mathcal{N}(0, \sigma^2 P_S)$. The term $-\eta \mu_F v$ represents a small, constant drift along the river direction, with $\mu_F > 0$. This captures the effect of a persistent, non-zero expected gradient component along the river, typical in late-stage training. We assume iterates remain in $U$.

**Assumption C.4** (Quadratic mountain component on $U$). On $U$, the loss decomposes as $L(w) = \ell(P_F(w - w_\star)) + \frac{1}{2}\|H^{1/2}P_S(w - w_\star)\|_2^2$, where $\ell : \mathbb{R} \to \mathbb{R}$ is $C^2$ and $H$ is a symmetric matrix satisfying $HP_F = 0$, $HP_S = P_S H = H$, and its spectrum on $\text{range}(P_S)$ is contained in $[\mu, L_S]$ for some $0 < \mu \leq L_S$.

**Decoupled dynamics.** The straight-river and quadratic-mountain assumptions imply that the SGD dynamics separate into a one-dimensional river recursion and a $(d - 1)$-dimensional mountain recursion. In particular, the river coordinate $t_k$ evolves without depending on the mountain state $z_k$, and the mountain recursion depends only on $z_k$ and the projected noise. Under Assumptions C.2–C.4, writing $t_k := v^\top(w_k - w_\star)$ and $z_k := P_S(w_k - w_\star)$, the updates satisfy

$$t_{k+1} = t_k - \eta \, \ell'(t_k) - \eta \mu_F, \qquad z_{k+1} = (I - \eta H)z_k + \eta g_k,$$

with $g_k \sim \mathcal{N}(0, \sigma^2 P_S)$.

**Checkpoint protocol.** We save checkpoints every $T$ steps: $w^{(m)} := w_{k_0 + mT}$ for $m = 0, 1, 2, \ldots$, starting after $k_0$.

## C.2. Theorem 1: Averaging Reduces Mountain Oscillations

We first formalize the distance to the river manifold. Under Assumption C.2, this distance is simply the norm of the projection onto the mountain subspace.

**Definition C.5** (River deviation). Under Assumption C.2, the river deviation of a point $w$ is $D(w) := \|P_S(w - w_\star)\|_2 = \text{dist}(w, M)$.

For the sake of readability, we restate Theorem 5.3 as below.

**Theorem C.6** (Averaging reduces expected distance to the river). *Assume Assumptions C.2–C.4. Let $\eta \in (0, 2/L_S)$ and let checkpoints $\{w^{(m)}\}_{m=0}^{N-1}$ be taken after a burn-in period $k_0$ such that the mountain dynamics are stationary. Let $\bar{w} = \frac{1}{N}\sum_{m=0}^{N-1} w^{(m)}$ be the merged checkpoint. Then the expected squared river deviation of $\bar{w}$ is*

$$\mathbb{E}\,D(\bar{w})^2 \;=\; \sum_{j=1}^{d-1}\left(\frac{\eta\sigma^2}{2\lambda_j - \eta\lambda_j^2}\right) \cdot \frac{1}{N^2}\left[N + 2\sum_{r=1}^{N-1}(N - r)\big(1 - \eta\lambda_j\big)^{rT}\right], \tag{5}$$

*where $\{\lambda_j\}_{j=1}^{d-1}$ are the eigenvalues of $H$ on $\text{range}(P_S)$. In particular, if checkpoints are weakly correlated such that $|1 - \eta\lambda_j|^T \leq \varepsilon$ for all $j$, then*

$$\mathbb{E}\,D(\bar{w})^2 \;\leq\; \frac{1}{N}\left(1 + \frac{2\varepsilon}{1 - \varepsilon}\right)\sum_{j=1}^{d-1}\frac{\eta\sigma^2}{2\lambda_j - \eta\lambda_j^2}. \tag{6}$$

*Proof.* Define the mountain coordinate $z_k := P_S(w_k - w_\star)$. We project the SGD update from Assumption C.3 onto the mountain subspace $\text{range}(P_S)$:

$$\begin{aligned}
z_{k+1} &= P_S(w_{k+1} - w_\star) \\
&= P_S\left((w_k - w_\star) - \eta\nabla L(w_k) - \eta\mu_F v + \eta g_k\right) \\
&= P_S(w_k - w_\star) - \eta P_S(\nabla L(w_k)) - \eta\mu_F P_S(v) + \eta P_S(g_k).
\end{aligned}$$

Under our assumptions:

- From Assumption C.4, $\nabla L(w) = \ell'(P_F(w-w_\star))v + HP_S(w-w_\star)$. Thus, $P_S(\nabla L(w_k)) = P_S(\ell'(\cdot)v) + P_S(Hz_k) = 0 + Hz_k = Hz_k$.

- From Assumption C.2, $P_S v = 0$.

- From Assumption C.3, $g_k$ is supported on $\mathrm{range}(P_S)$, so $P_S g_k = g_k$.

Substituting these into the update of $z_k$:

$$z_{k+1} = z_k - \eta H z_k + \eta g_k = (I - \eta H)z_k + \eta g_k.$$

Let $A := I - \eta H$. The condition $\eta \in (0, 2/L_S)$ ensures that the eigenvalues of $A$ on $\mathrm{range}(P_S)$ are in $(-1, 1)$.

Let $\{q_j\}_{j=1}^{d-1}$ be an orthonormal eigenbasis for $H$ on $\mathrm{range}(P_S)$ with eigenvalues $\lambda_j$. Let $u_k^{(j)} := q_j^\top z_k$ be the coordinate of $z_k$ along $q_j$. Projecting the dynamics of $z_k$ onto $q_j$:

$$
\begin{aligned}
u_{k+1}^{(j)} = q_j^\top z_{k+1} &= q_j^\top \left((I - \eta H)z_k + \eta g_k\right) \\
&= q_j^\top (I - \eta H)z_k + \eta q_j^\top g_k \\
&= (1 - \eta\lambda_j)q_j^\top z_k + \eta q_j^\top g_k \\
&= (1 - \eta\lambda_j)u_k^{(j)} + \eta\xi_k^{(j)},
\end{aligned}
$$

where $\xi_k^{(j)} := q_j^\top g_k$. Since $g_k \sim \mathcal{N}(0, \sigma^2 P_S)$ and $\{q_j\}$ is an orthonormal basis for $\mathrm{range}(P_S)$, the variables $\xi_k^{(j)}$ are i.i.d. $\mathcal{N}(0, \sigma^2)$. This is an AR(1) process[6] with coefficient $a_j := 1 - \eta\lambda_j \in (-1, 1)$.

For an AR(1) process $x_{t+1} = ax_t + \epsilon_t$ with i.i.d. noise $\epsilon_t$ where $\mathbb{E}[\epsilon_t] = 0$, $\mathrm{Var}(\epsilon_t) = \sigma_\epsilon^2$, the stationary variance is $\mathrm{Var}(x) = \sigma_\epsilon^2/(1-a^2)$. For our process $u_k^{(j)}$, the noise term is $\eta\xi_k^{(j)}$, so its variance is $\mathrm{Var}(\eta\xi_k^{(j)}) = \eta^2\sigma^2$.

$$\mathrm{Var}(u_k^{(j)}) = \frac{\eta^2\sigma^2}{1-a_j^2} = \frac{\eta^2\sigma^2}{1-(1-\eta\lambda_j)^2} = \frac{\eta^2\sigma^2}{1-(1-2\eta\lambda_j+\eta^2\lambda_j^2)} = \frac{\eta\sigma^2}{2\lambda_j - \eta\lambda_j^2}.$$

The autocovariance at lag $\Delta \geq 0$ is $\mathrm{Cov}(u_k^{(j)}, u_{k+\Delta}^{(j)}) = a_j^\Delta \mathrm{Var}(u_k^{(j)})$.

Now we formalize the variance of the averaged checkpoint. Let $\bar{z} = \frac{1}{N}\sum_{m=0}^{N-1} z^{(m)}$, where $z^{(m)} = z_{k_0+mT}$. The expected squared deviation is $\mathbb{E}D(\bar{w})^2 = \mathbb{E}\|\bar{z}\|_2^2$. By orthogonality of the basis $\{q_j\}$, this is $\sum_{j=1}^{d-1}\mathbb{E}[(\bar{u}^{(j)})^2]$. Since the process is stationary and zero-mean, $\mathbb{E}[(\bar{u}^{(j)})^2] = \mathrm{Var}(\bar{u}^{(j)})$.

$$
\begin{aligned}
\mathrm{Var}(\bar{u}^{(j)}) = \mathrm{Var}\left(\frac{1}{N}\sum_{m=0}^{N-1} u_{k_0+mT}^{(j)}\right) &= \frac{1}{N^2}\sum_{m=0}^{N-1}\sum_{n=0}^{N-1}\mathrm{Cov}(u_{k_0+mT}^{(j)}, u_{k_0+nT}^{(j)}) \\
&= \frac{1}{N^2}\sum_{m=0}^{N-1}\sum_{n=0}^{N-1} a_j^{|m-n|T}\mathrm{Var}(u^{(j)}) \\
&= \frac{\mathrm{Var}(u^{(j)})}{N^2}\left(\sum_{m=n} a_j^{0\cdot T} + \sum_{m\neq n} a_j^{|m-n|T}\right).
\end{aligned}
$$

The double summation can be re-indexed by the lag $r = |m - n|$.

- For lag $r = 0$ (diagonal terms, $m = n$), there are $N$ terms, each equal to $a_j^0 = 1$.

- For lag $r \in \{1, \ldots, N-1\}$, there are $2(N-r)$ pairs $(m, n)$ such that $|m - n| = r$.

---

[6] AutoRegressive model of order 1

Thus, the sum becomes:

$$\mathrm{Var}(\bar{u}^{(j)}) = \frac{\mathrm{Var}(u^{(j)})}{N^2}\left(N + 2\sum_{r=1}^{N-1}(N-r)a_j^{rT}\right).$$

Summing over $j$ and substituting the expression for $\mathrm{Var}(u^{(j)})$ gives the first result. Then, we use the triangle inequality and the sum of a geometric series. Note that $|a_j| = |1 - \eta\lambda_j| < 1$.

$$\frac{1}{N^2}\left|N + 2\sum_{r=1}^{N-1}(N-r)a_j^{rT}\right| \le \frac{1}{N^2}\left(N + 2\sum_{r=1}^{N-1}(N-r)|a_j|^{rT}\right)$$

$$\le \frac{1}{N^2}\left(N + 2(N-1)\sum_{r=1}^{N-1}(|a_j|^T)^r\right)$$

$$\le \frac{1}{N^2}\left(N + 2(N-1)\sum_{r=1}^{\infty}(|a_j|^T)^r\right)$$

$$= \frac{1}{N^2}\left(N + 2(N-1)\frac{|a_j|^T}{1-|a_j|^T}\right).$$

Using the condition $|a_j|^T \le \varepsilon$:

$$\mathrm{Var}(\bar{u}^{(j)}) \le \frac{\mathrm{Var}(u^{(j)})}{N^2}\left(N + 2(N-1)\frac{\varepsilon}{1-\varepsilon}\right)$$

$$= \mathrm{Var}(u^{(j)})\left(\frac{1}{N} + \frac{2(N-1)}{N^2}\frac{\varepsilon}{1-\varepsilon}\right).$$

$$\le \frac{1}{N}\left(1 + \frac{2\varepsilon}{1-\varepsilon}\right)\mathrm{Var}(u^{(j)}).$$

Summing over $j$ yields the second result. $\qquad\square$

## C.3. Theorem 2: PCA on Sliding-Window Merges Recovers the River Direction

We analyze a sequence of $K$ merged checkpoints from sliding windows: $\bar{w}^{(s)} := \frac{1}{N}\sum_{i=0}^{N-1}w^{(s+i)}$. We perform PCA on the centered sequence $\tilde{w}^{(s)} := \bar{w}^{(s)} - \frac{1}{K}\sum_{r=0}^{K-1}\bar{w}^{(r)}$. For the sake of readability, we restate Theorem 5.4 in a slightly more detailed form as below.

**Theorem C.7** (PCA direction aligns with the flattest eigenvector). *Assume Assumptions C.2–C.4. Decompose each merged state as $\bar{w}^{(s)} = w_\star + t_s v + \varepsilon_s$. Let $\Sigma = \mathbb{E}[\tilde{w}^{(s)}\tilde{w}^{(s)\top}]$ be the population covariance. Let $\sigma_{\mathrm{sig}}^2 := \mathrm{Var}(\tilde{t}_s)$ and $\sigma_{\mathrm{noise}}^2 := \lambda_{\max}(\mathbb{E}[\tilde{\varepsilon}_s\tilde{\varepsilon}_s^\top])$. If the signal-to-noise gap $\delta_\Sigma := \sigma_{\mathrm{sig}}^2 - \sigma_{\mathrm{noise}}^2 > 0$, then $v$ is the unique top eigenvector of $\Sigma$.*

*Furthermore, under a standard concentration assumption for the sample covariance $\hat{\Sigma}$, with probability at least $1 - \alpha$:*

$$|\hat{u}_1^\top v|^2 \ge 1 - \left(\frac{\|\hat{\Sigma} - \Sigma\|_{\mathrm{op}}}{\delta_\Sigma}\right)^2, \tag{7}$$

*where $\hat{u}_1$ is the top eigenvector of $\hat{\Sigma}$.*

*Proof.* The centered merged state is $\tilde{w}^{(s)} = \tilde{t}_s v + \tilde{\varepsilon}_s$, where $\tilde{t}_s$ and $\tilde{\varepsilon}_s$ are the centered versions of $t_s$ and $\varepsilon_s$. The population covariance matrix is:

$$\Sigma = \mathbb{E}[\tilde{w}^{(s)}(\tilde{w}^{(s)})^\top] = \mathbb{E}\left[(\tilde{t}_s v + \tilde{\varepsilon}_s)(\tilde{t}_s v + \tilde{\varepsilon}_s)^\top\right]$$

$$= \mathbb{E}\left[\tilde{t}_s^2 vv^\top + \tilde{t}_s v\tilde{\varepsilon}_s^\top + \tilde{\varepsilon}_s\tilde{t}_s v^\top + \tilde{\varepsilon}_s\tilde{\varepsilon}_s^\top\right]$$

$$= \mathbb{E}[\tilde{t}_s^2]vv^\top + \mathbb{E}[\tilde{t}_s v\tilde{\varepsilon}_s^\top] + \mathbb{E}[\tilde{\varepsilon}_s\tilde{t}_s v^\top] + \mathbb{E}[\tilde{\varepsilon}_s\tilde{\varepsilon}_s^\top].$$

By construction, $\varepsilon_s := P_S(\bar{w}^{(s)} - w_\star)$ lies in the mountain subspace $\mathrm{range}(P_S)$, hence it is orthogonal to $v$, i.e., $v^\top \varepsilon_s = 0$. Centering is a linear operation, so $\tilde{\varepsilon}_s$ also lies in $\mathrm{range}(P_S)$ and satisfies $v^\top \tilde{\varepsilon}_s = 0$. Therefore, since $\tilde{t}_s$ and $\tilde{\varepsilon}_s$ are independent:

$$\mathbb{E}[\tilde{t}_s v \tilde{\varepsilon}_s^\top] = v\mathbb{E}[\tilde{t}_s \tilde{\varepsilon}_s^\top] = v\mathbb{E}[\tilde{t}_s]\mathbb{E}[\tilde{\varepsilon}_s^\top] = v \cdot 0 \cdot \mathbf{0}^\top = \mathbf{0}.$$

Then the covariance matrix simplifies to a block-diagonal structure:

$$\Sigma = \mathbb{E}[\tilde{t}_s^2]vv^\top + \mathbb{E}[\tilde{\varepsilon}_s \tilde{\varepsilon}_s^\top].$$

Since $\mathbb{E}[\tilde{t}_s] = 0$, we have $\mathbb{E}[\tilde{t}_s^2] = \mathrm{Var}(\tilde{t}_s) =: \sigma_{\mathrm{sig}}^2$. Let $\Sigma_{\varepsilon,\mathrm{cent}} := \mathbb{E}[\tilde{\varepsilon}_s \tilde{\varepsilon}_s^\top]$. The covariance matrix is:

$$\Sigma = \sigma_{\mathrm{sig}}^2 vv^\top + \Sigma_{\varepsilon,\mathrm{cent}}.$$

The noise component $\varepsilon_s = P_S(\bar{w}^{(s)} - w_\star)$ lies in the mountain subspace $\mathrm{range}(P_S)$, which is orthogonal to $v$. The centered noise $\tilde{\varepsilon}_s$ also lies in this subspace. Therefore, the covariance $\Sigma_{\varepsilon,\mathrm{cent}}$ is supported on $\mathrm{range}(P_S)$, which implies $\Sigma_{\varepsilon,\mathrm{cent}} v = \mathbf{0}$. Let's test $v$ as an eigenvector of $\Sigma$:

$$\Sigma v = (\sigma_{\mathrm{sig}}^2 vv^\top + \Sigma_{\varepsilon,\mathrm{cent}})v = \sigma_{\mathrm{sig}}^2 v(v^\top v) + \Sigma_{\varepsilon,\mathrm{cent}}v = \sigma_{\mathrm{sig}}^2 v(1) + \mathbf{0} = \sigma_{\mathrm{sig}}^2 v.$$

So, $v$ is an eigenvector with eigenvalue $\sigma_{\mathrm{sig}}^2$. Now consider any unit vector $u \perp v$. Such a vector $u$ lies in $\mathrm{range}(P_S)$.

$$\Sigma u = (\sigma_{\mathrm{sig}}^2 vv^\top + \Sigma_{\varepsilon,\mathrm{cent}})u = \sigma_{\mathrm{sig}}^2 v(v^\top u) + \Sigma_{\varepsilon,\mathrm{cent}}u = \sigma_{\mathrm{sig}}^2 v(0) + \Sigma_{\varepsilon,\mathrm{cent}}u = \Sigma_{\varepsilon,\mathrm{cent}}u.$$

This shows that the action of $\Sigma$ on the subspace orthogonal to $v$ is identical to the action of $\Sigma_{\varepsilon,\mathrm{cent}}$. The eigenvalues of $\Sigma$ are therefore $\{\sigma_{\mathrm{sig}}^2\} \cup \{\text{eigenvalues of } \Sigma_{\varepsilon,\mathrm{cent}}\}$. The largest eigenvalue of $\Sigma_{\varepsilon,\mathrm{cent}}$ is $\lambda_{\max}(\Sigma_{\varepsilon,\mathrm{cent}}) =: \sigma_{\mathrm{noise}}^2$. Thus, the eigenvalues of $\Sigma$ are $\lambda_1(\Sigma) = \sigma_{\mathrm{sig}}^2$ and $\lambda_2(\Sigma) = \sigma_{\mathrm{noise}}^2$ (assuming no other eigenvalue of $\Sigma_{\varepsilon,\mathrm{cent}}$ is larger). The condition $\delta_\Sigma = \sigma_{\mathrm{sig}}^2 - \sigma_{\mathrm{noise}}^2 > 0$ is precisely the condition that $\lambda_1(\Sigma) > \lambda_2(\Sigma)$. By the properties of symmetric matrices, this makes the top eigenvector $v$ unique. Then using Davis-Kahan $\sin\Theta$ theorem, we have:

$$\sin\angle(\hat{u}_1, v) \leq \frac{\|\hat{\Sigma} - \Sigma\|_{\mathrm{op}}}{\lambda_1(\Sigma) - \lambda_2(\Sigma)}.$$

In our notation, the eigengap is $\lambda_1(\Sigma) - \lambda_2(\Sigma) = \sigma_{\mathrm{sig}}^2 - \sigma_{\mathrm{noise}}^2 = \delta_\Sigma$. So,

$$\sin\angle(\hat{u}_1, v) \leq \frac{\|\hat{\Sigma} - \Sigma\|_{\mathrm{op}}}{\delta_\Sigma}.$$

The squared cosine of the angle is what we need:

$$|\hat{u}_1^\top v|^2 = \cos^2 \angle(\hat{u}_1, v) = 1 - \sin^2 \angle(\hat{u}_1, v) \geq 1 - \left(\frac{\|\hat{\Sigma} - \Sigma\|_{\mathrm{op}}}{\delta_\Sigma}\right)^2.$$

This completes the proof. $\qquad\square$

**Analysis of the Signal-to-Noise Gap.** The condition $\delta_\Sigma > 0$ is crucial. We now connect $\sigma_{\mathrm{sig}}^2$ and $\sigma_{\mathrm{noise}}^2$ to the hyperparameters $(N, T, K)$.

**Lemma C.8** (Residual mountain energy). *Under the conditions of Theorem C.6, the residual noise variance is bounded by:*
$\sigma_{\mathrm{noise}}^2 = \lambda_{\max}(\Sigma_{\varepsilon,\mathrm{cent}}) \leq \mathrm{tr}(\Sigma_{\varepsilon,\mathrm{cent}}) \leq \mathbb{E}\|\varepsilon_s\|_2^2 = \mathbb{E}D(\bar{w}^{(s)})^2$. *The RHS is given by the expression in Theorem C.6.*

*Proof.* For any positive semidefinite matrix $A$, its largest eigenvalue (operator norm) is bounded by its trace: $\lambda_{\max}(A) \leq \mathrm{tr}(A)$. Applying this to $\Sigma_{\varepsilon,\mathrm{cent}}$:

$$\sigma_{\mathrm{noise}}^2 = \lambda_{\max}(\Sigma_{\varepsilon,\mathrm{cent}}) \leq \mathrm{tr}(\Sigma_{\varepsilon,\mathrm{cent}}).$$

The trace is the sum of diagonal elements, which equals the expected squared Frobenius norm:

$$\mathrm{tr}(\Sigma_{\varepsilon,\mathrm{cent}}) = \mathrm{tr}(\mathbb{E}[\tilde{\varepsilon}_s \tilde{\varepsilon}_s^\top]) = \mathbb{E}[\mathrm{tr}(\tilde{\varepsilon}_s \tilde{\varepsilon}_s^\top)] = \mathbb{E}[\|\tilde{\varepsilon}_s\|_2^2].$$

The centered vector $\tilde{\varepsilon}_s = \varepsilon_s - \frac{1}{K}\sum_{r=0}^{K-1}\varepsilon_r$. The sample mean $\bar{\varepsilon} = \frac{1}{K}\sum_r \varepsilon_r$ is the vector that minimizes the sum of squared distances $\sum_s \|\varepsilon_s - c\|_2^2$. Therefore, $\mathbb{E}[\|\varepsilon_s - \bar{\varepsilon}\|_2^2] \leq \mathbb{E}[\|\varepsilon_s - c\|_2^2]$ for any constant vector $c$. Since the mountain process is stationary and zero-mean, $\mathbb{E}[\varepsilon_s] = 0$. Choosing $c = 0$ gives:

$$\mathbb{E}[\|\tilde{\varepsilon}_s\|_2^2] \leq \mathbb{E}[\|\varepsilon_s\|_2^2].$$

Finally, by definition, $\varepsilon_s = P_S(\bar{w}^{(s)} - w_\star)$, so its squared norm is the squared river deviation:

$$\mathbb{E}[\|\varepsilon_s\|_2^2] = \mathbb{E}[\|P_S(\bar{w}^{(s)} - w_\star)\|_2^2] = \mathbb{E}D(\bar{w}^{(s)})^2.$$

Combining these inequalities gives the desired result. $\qquad\square$

**Lemma C.9** (Signal variance from river drift). *Under Assumption C.3, let $\ell'(t) \approx \ell_0'$ be the nearly constant river-gradient component in $U$. The expected progress of a merged checkpoint per sliding step is approximately $\mathbb{E}[t_{s+1} - t_s] \approx -T\eta(\ell_0' + \mu_F)$. We define the magnitude of this step as $\Delta_t := T\eta(\ell_0' + \mu_F)$. The signal variance is then lower-bounded by: $\sigma_{\text{sig}}^2 = \text{Var}(\tilde{t}_s) \geq \Delta_t^2 \cdot \frac{K^2-1}{12}$.*

*Proof.* The river dynamics are $t_{k+1} = t_k - \eta\ell'(t_k) - \eta\mu_F$. With the approximation $\ell'(t_k) \approx \ell_0'$, this becomes an arithmetic progression: $t_{k+1} \approx t_k - \eta(\ell_0' + \mu_F)$. The position of the $s$-th merged checkpoint is $t_s = \frac{1}{N}\sum_{i=0}^{N-1} t_{k_0+(s+i)T}$. Its expectation is:

$$\mathbb{E}[t_s] \approx \frac{1}{N}\sum_{i=0}^{N-1}(\mathbb{E}[t_{k_0}] - (s+i)T\eta(\ell_0' + \mu_F)) = C - s \cdot T\eta(\ell_0' + \mu_F) = C - s\Delta_t.$$

The sequence of expectations $\{\mathbb{E}[t_s]\}_{s=0}^{K-1}$ is an arithmetic progression. The variance of the centered process, $\sigma_{\text{sig}}^2 = \text{Var}(\tilde{t}_s)$, is lower-bounded by the variance of the centered means (by the Law of Total Variance):

$$\sigma_{\text{sig}}^2 = \text{Var}(\tilde{t}_s) \geq \text{Var}(\mathbb{E}[\tilde{t}_s]) = \text{Var}(\mathbb{E}[t_s] - \frac{1}{K}\sum_r \mathbb{E}[t_r]).$$

This is the sample variance of the deterministic sequence $\{C - s\Delta_t\}_{s=0}^{K-1}$. The constant $C$ drops out, and we compute the variance of $\{-s\Delta_t\}_{s=0}^{K-1}$:

$$\text{Var}(-s\Delta_t) = \Delta_t^2\text{Var}(s) = \Delta_t^2\left(\frac{1}{K}\sum_{s=0}^{K-1}s^2 - \left(\frac{1}{K}\sum_{s=0}^{K-1}s\right)^2\right)$$

$$= \Delta_t^2\left(\frac{(K-1)(2K-1)}{6} - \left(\frac{K-1}{2}\right)^2\right)$$

$$= \Delta_t^2\left(\frac{2(K-1)(2K-1) - 3(K-1)^2}{12}\right)$$

$$= \Delta_t^2\frac{(K-1)(4K-2-3K+3)}{12}$$

$$= \Delta_t^2\frac{(K-1)(K+1)}{12} = \Delta_t^2\frac{K^2-1}{12}.$$

This provides the lower bound for the signal variance. $\qquad\square$

Combining these lemmas, a sufficient condition for $\delta_\Sigma > 0$ is:

$$\underbrace{(T\eta(\ell_0' + \mu_F))^2 \cdot \frac{K^2-1}{12}}_{\text{Signal (LHS)}} > \underbrace{\sum_{j=1}^{d-1}\left(\frac{\eta\sigma^2}{2\lambda_j - \eta\lambda_j^2}\right) \cdot \frac{1}{N^2}\left[N + 2\sum_{r=1}^{N-1}(N-r)(1-\eta\lambda_j)^{rT}\right]}_{\text{Noise (RHS)}}.$$

This inequality clearly shows how the hyperparameters control the signal-to-noise gap:

- Increasing $N$ or $T$ suppresses the noise term on the RHS.

- Increasing $K$ or the drift-per-step $\Delta_t$ (via $T$) amplifies the signal term on the LHS.

