# OpenReview forum: "Extra-Merge: Tracing the Rank-1 Subspace of Model Merging in Language Model Pre-Training"
_ICML.cc/2026/Conference — ICML 2026 regular_

### Official Review · Reviewer_EPBi · 2026-03-10

**Soundness:** 2
**Presentation:** 1
**Significance:** 1
**Originality:** 1
**Overall Recommendation:** 2
**Confidence:** 5

**Summary:**

This paper introduces Extra-Merge, a training-free strategy designed to enhance Large Language Model pre-training by exploiting the geometric properties of model merging trajectories. The authors observe a Rank-1 Subspace phenomenon, where merged checkpoints collapse onto a stable, nearly one-dimensional linear manifold, unlike raw optimization steps that oscillate violently. By performing PCA on a sliding window of merged checkpoints and extrapolating along the first principal component, the method seeks to find lower-loss solutions without additional gradient updates.

**Compliance With Llm Reviewing Policy:**

Affirmed.

**Ethical Review Concerns:**

Nothing.

**Final Justification:**

I have read the author's response. The author's findings may be novel, but this does not convince me of my original view on the inherent limitations of the overall work. I believe the contribution is incremental and may require more thorough revisions.

**Key Questions For Authors:**

1. What is the peak memory of applying Extre-Merge?
2. How does it perform on the tasks that were not listed in Table 1?

**Limitations:**

yes

**Strengths And Weaknesses:**

# Strengths:
The paper tries to address the critical challenge of reducing the immense computational costs of LLM pre-training by exploring "free-lunch" performance gains through post-hoc model merging.

# Weaknesses:
## 1. Oversimplified Theoretical Foundation and Limited Novelty
The theoretical framework of the paper relies heavily on the River-Valley landscape perspective introduced by WSD [1] to justify the observed Rank-1 subspace phenomenon. However, the paper suffers from significant issues regarding the strength of its assumptions and the originality of the resulting algorithm. The core of the theoretical proof depends on Assumption C.2, which posits that the descent direction $v(w) \equiv v$ is constant throughout the neighborhood. This is a substantially tighter and more restrictive condition than Assumption 2.6 in the original WSD. By assuming a perfectly straight descent path, the complex non-convex optimization is reduced to a simple process. Under such a simplified model, proving that averaging reduces variance and that the trajectory collapses onto a one-dimensional manifold becomes a basic result in stochastic processes rather than a significant contribution to optimization.

## 2. Circular Reasoning in Rank-1 Discovery
There is an inherent logical circularity in the paper’s discussion. The authors use a theoretical model that assumes a Straight River to provide evidence of a Rank-1 Subspace. Because the Rank-1 property is naturally baked into the assumptions,  the empirical findings in Section 4.2 feel more like a verification of the WSD scheduler’s linear decay phase rather than an emergent property of LLM landscapes themselves.

## 3. Poor Presentation
There are too many typos in the presentation. For example, in lines 134 and 135, in the right column, it is possible that the hyphens were omitted when writing the formula. Similar typos happen in lines 245 and 259.  Some symbols also lack definitions, like in line 316, what is M. The author should carefully review the manuscript before submission to avoid many expressions that do not match the paper.

## 4. Engineering and Practicality Concerns:
The method faces a significant engineering bottleneck. It requires loading multiple checkpoints and calculating full-parameter inner products, which is extremely memory-intensive. This implies that applying the method even to smaller models could incur memory costs comparable to larger models. The authors should have reported the peak memory usage. Furthermore, as shown in the results, the improvements in both training loss and downstream tasks are limited, making the high memory overhead an unacceptable bottleneck.

[1] Kaiyue Wen, Zhiyuan Li, Jason Wang, David Hall, Percy Liang, Tengyu Ma, "Understanding Warmup-Stable-Decay Learning Rates: A River Valley Loss Landscape Perspective"

---

> ### Author Rebuttal · Authors · 2026-03-28
>
> We sincerely thank the reviewer for the feedback. We address each concern below and hope our responses clarify the raised points.
>
> **W1: Oversimplified Theoretical Foundation and Limited Novelty**
>
> **A1.1 (Originality).** We respectfully hold a different view on this matter.
> **(i)** The Rank-1 ridge is a **brand-new discovery**: experiments on GPT-2 S/M and LLaMA-0.5B/2B reveal for the first time that *merged* checkpoints exhibit rank-1 structure while *raw* ones do not, this is unexplored by Wen et al. or subsequent work. Critically, it appears in both WSD's non-decay phase and cosine schedules, and is thus unrelated to LR decay. The Rank-1 ridge constitutes a novel LLM landscape property.
>
> **(ii)** Extra-Merge derives **directly from the Rank-1 ridge**, not the river-valley perspective. Its core is extrapolation along a loss-decreasing direction among merged models. As stated in line 286, the river-valley framework serves solely as *theoretical insight* to explain the mechanism, it plays no role in algorithm derivation.
>
> **A1.2 (Assumption C.2).** We respectfully disagree on several grounds. **(a)** Comparing C.2 with WSD's Assumption 2.6 is inaccurate; they address different properties. C.2's proper counterpart is the Straight River assumption that WSD likewise adopts in its stochastic analysis, not the gradient-flow conservation in 2.6. **(b)** C.2 does *not* assume a straight optimization path; it only assumes the local flattest direction is approximately constant. Raw trajectories still oscillate in the $(d_M{-}1)$-dimensional mountain subspace; rank-1 structure emerges only after merging suppresses these oscillations, it is not assumed. **(c)** "Averaging reduces variance" does *not* automatically yield rank-1 collapse. Section 4.2 provides a counterexample: for an isotropic random walk, sliding-window averaging shrinks covariance uniformly *without* producing a dominant PC1. Our result relies non-trivially on anisotropy, temporal correlation, and persistent drift. **(d)** Section 5.1 is labeled *Theoretical Insight*, only providing mechanistic explanation and actionable predictions absent from Wen et al. and PMA/LAWA.
>
> **W2: Circular Reasoning in Rank-1 Discovery**
>
> **A2.** We respectfully disagree that circular reasoning is present.
>
> **(1)** Our logic is: empirical discovery first, theoretical explanation second, not "assume rank-1, verify rank-1." The Straight River assumption describes *local loss geometry* (the direction of lowest curvature) and does not entail that the checkpoint trajectory is rank-1. Under it, raw dynamics $z_{k+1}=(I-\eta H)z_k+\eta g_k$ retain high-dimensional mountain noise, explaining why raw checkpoints are *not* rank-1 while merged ones are. Theorem 5.4 further requires an SNR condition $\rho>1$ for PCA to recover the river direction; this would be superfluous if rank-1 were already assumed.
>
> **(2)** Section 4.2 is not a re-verification of WSD's decay phase. On the same GPT-2/WSD trajectory, raw and merged checkpoints share the identical schedule yet differ markedly in geometry (Figs. 2–3): raw ≠ rank-1, merged = rank-1. The decisive factor is merging's restructuring of trajectory geometry, not the schedule. Fig. 4 shows rank-1 stability strengthens systematically with $N$ while the schedule is fixed. Our LLaMA + cosine LR experiments (Fig. 3) reproduce the same phenomenon, confirming it is schedule-independent.
>
> **W3: Poor Presentation**
>
> **A3.** We sincerely thank the reviewer for catching these issues. We will correct all three errors (lines 134–135, 245, 259) and deferred symbol $M$ (line 316) defined in the appendix. The revision will be carefully proofread to prevent such issues.
>
> **W4 & KQ1: Peak Memory and Engineering Concerns**
>
> **A4.** We thank the reviewer for raising this concern, but kindly disagree that Extra-Merge constitutes a significant engineering bottleneck. Appendix B.5 detailed the direction computation; to further clarify, we expanded the full stage-by-stage breakdown of peak memory and wall-clock time in **our response to Reviewer mCjZ (W3)**, to which we kindly refer the reviewer due to space constraints. In summary: Extra-Merge uses the **Gram-matrix trick** with **shard-wise streaming**, requiring peak CPU RAM that scales **linearly with model size** (33.5 GB at 12B, reducible via finer-grained streaming), and the entire CPU-only pipeline takes **~4 min for Pythia-12B**—negligible vs. training cost. We sincerely hope this clarification alleviates the concern.
>
> **KQ2: Tasks not in Table 1**
>
> We thank the reviewer for this question. To comprehensively demonstrate Extra-Merge's downstream gains, we have conducted two additional experiments: extended benchmarks for Pythia-12B and a 10-task evaluation for OLMo-7B. Due to space limits, we kindly refer the reviewer to **Table 1 and Table 2 in our response to Reviewer zoPw**. We hope these results address your question; please let us know if further clarification is needed.

---

> > ### Author Rebuttal · Reviewer_EPBi · 2026-04-03
> >
> > I would like to thank the authors for their detailed rebuttal. To some extent, it has clarified several of my concerns regarding the theoretical claims and the experimental. However, despite these clarifications, I maintain that the manuscript possesses inherent limitations that prevent me from upgrading my score. From a theoretical perspective, the work does not appear to introduce significantly new analytical techniques. While the observations may be considered interesting by some, the underlying intuition remains somewhat straightforward. Specifically, the idea of averaging multiple local solutions in a non-convex landscape is not newl for optimization analysis. For instance, analysis for SGD on non-convex settings [1] has already considered to use averaging points for better convergence. Regarding the empirical results, the performance gains provided by the proposed algorithm appear limited. The improvement in loss is on the order of 0.01, and the gains on downstream tasks are approximately 1%. Without a more substantial performance gap or a demonstration that the baseline has been pushed to its absolute limit, it is difficult to conclude that the proposed approach offers a significant practical advantage.
> >
> > [1] Sharp Analysis for Nonconvex SGD Escaping from Saddle Points

---

> > > ### Author Response · Authors · 2026-04-03
> > >
> > > We sincerely thank the reviewer for acknowledging that our rebuttal has *clarified their concerns* raised in the original review. We would like to take this opportunity to briefly re-emphasize our contributions and respectfully address what we believe may be a remaining misunderstanding.
> > >
> > > **Our contributions form a complete empirical-to-algorithmic pipeline.** As recognized by Reviewers mCjZ and zoPw, the paper builds a *"full chain from empirical diagnosis to geometric interpretation, theoretical explanation, algorithm design, and experimental validation"* (Reviewer mCjZ), with *"clear figures, clear motivation, and a clear method description"* and findings that *"seem novel"* (Reviewer zoPw). Concretely:
> > >
> > > - **Contribution 1 (Discovery + Insight):** We empirically uncover, for the first time, that merged LLM checkpoints collapse onto a **Rank-1 Subspace**: a locally one-dimensional manifold, while raw checkpoints do not. We use the river-valley theoretical framework to provide *mechanistic insight*, not the discovery itself.
> > > - **Contribution 2 (Algorithm + Justification):** Capitalizing on this geometric property, we propose **Extra-Merge**, a training-free extrapolation strategy with theoretical guarantees on direction recovery (Theorem 5.4).
> > >
> > > **We respectfully note a possible misunderstanding regarding our claimed novelty.** The reviewer's comment and the cited reference ([1], nonconvex SGD analysis) concern the well-known idea that *averaging iterates improves convergence*. We fully agree this is classical and have never claimed it as our contribution—indeed, PMA/LAWA (our baselines) already embody this idea. Our novelty lies *beyond* averaging: it is the discovery that the *post-averaging* trajectory exhibits a stable rank-1 geometric structure, and that this structure can be *exploited via extrapolation* (not interpolation) to obtain further gains. This is a fundamentally different insight from "averaging helps," and we believe the distinction is important.
> > >
> > > **On the magnitude of improvements.** We respectfully note that Extra-Merge is compared against the current SOTA training-free baseline (PMA). As shown in our extended experiments (Tables 1–2 in our response to Reviewer zoPw), Extra-Merge's average downstream gain is **4–7× that of PMA**—a substantial margin for a method that requires zero additional training tokens. We believe this demonstrates clear practical value of the Rank-1 discovery.
> > >
> > > We hope this clarification helps resolve the remaining concerns. We are grateful for the reviewer's time and engagement throughout the discussion.

---

### Official Review · Reviewer_zoPw · 2026-03-11

**Soundness:** 3
**Presentation:** 4
**Significance:** 3
**Originality:** 3
**Overall Recommendation:** 5
**Confidence:** 4

**Summary:**

The paper focuses on checkpoint merging techniques in late-stage LLM pretraining. It starts with an analysis of training trajectories in the late stage and finds that consecutively merged checkpoints collapse onto an almost one-dimensional linear manifold, supported by solid empirical observations and theoretical insights. Based on this finding, the paper proposes the Extra-Merge technique, which leverages this Rank-1 Subspace and extrapolates the optimization trajectory to move toward an undiscovered low-loss region. Experiments are conducted by pretraining a GPT series of models with 10B tokens, as well as LLaMA models up to the 2B scale with 100B tokens to demonstrate improved training perplexity. The paper also uses existing model checkpoints and compares the downstream performance of several model merging techniques.

**Compliance With Llm Reviewing Policy:**

Affirmed.

**Final Justification:**

I think this is a good paper including the interesting finding, theory justification, sound method and correct experimental setup. My concerns are addressed and I would like to keep my score. I've also read other reviewers' comments.

**Key Questions For Authors:**

Please see above.

**Limitations:**

yes

**Strengths And Weaknesses:**

Soundness
1. The proposed method is motivated by the finding of the Rank-1 Subspace of merged checkpoints in late training. To support the motivation, the authors show strong empirical findings and theoretical insights, which look solid.

2. I checked the experimental setup in the Appendix and found that it is solid and fair. Experiments start from GPT-series pretraining across different model sizes with a small number of tokens, followed by large-scale training on 100B tokens, and finally show some downstream performance from existing checkpoints of a 12B model.

3. One problem I see is about the experimental results. The final downstream improvement is not very significant, with only a 0–1 point accuracy increase. While this can be affected by the setup, tasks, and model selections, it would be helpful if the authors could add more comparisons. For example, the authors could try more difficult tasks, such as GSM8K, to see whether there could be a larger gap. In addition, Pythia is a bit outdated, and more recent models such as Olmo provide much better downstream results. Could the authors also report checkpoint merging results with Olmo? I think they also provide intermediate checkpoints.

4. Another ablation that I think is important is about the number of merged checkpoints. How would this affect the performance of Extra-merge? Is it robust to different numbers of merged models?

Presentation

1. The presentation is good, with clear figures, clear motivation, and a clear method description. The structure of the different sections is also reasonable.

2. Given that this paper contains many proofs, I appreciate that the authors provide intuition and necessary summaries before diving into the mathematical details. I think this makes reading and understanding much easier.

Significance

1. I think the paper focuses on an important problem: effectively using different checkpoints during large-scale LLM training to boost the final performance and achieve more stable results. Given the long training time and high cost of training advanced LLMs, exploring checkpoint merging is significant.

Originality

1. The finding of the Rank-1 Subspace seems novel. The proposed method is also new and novel while being simple and well aligned with the findings.

2. Besides the AdamW optimizer, the paper also validates the effectiveness of the proposed method with the Muon optimizer.

---

> ### Author Rebuttal · Authors · 2026-03-28
>
> We sincerely thank the reviewer for recognizing the solidity of our empirical findings and theoretical insights, as well as the fairness of our experimental setup. We address each concern below.
>
> **S3.1: More comparisons on downstream tasks, including harder benchmarks such as GSM8K**
>
> A3.1 : We sincerely thank the reviewer for this valuable suggestion. Following your suggestion, we additionally evaluate Pythia-12B on six benchmarks beyond the original four: Winogrande, BoolQ, OBQA, GSM8K, TruthfulQA-MC2, and MMLU, using lm-eval (v0.4.10.dev0, bf16). All tasks are zero-shot except GSM8K (5-shot, greedy). Metrics: accuracy for Winogrande/BoolQ/MMLU, acc\_norm for OBQA, exact\_match for GSM8K, MC2 for TruthfulQA. EM uses default settings (K=4, N=10). Pile validation PPL is included to show our α-selection criterion.
>
> **Table 1.** Extended Pythia-12B results.
>
> | Model | PPL↓ | Wino. | BoolQ | OBQA | GSM8K | TQA | MMLU | Avg. |
> |:---|:---:|:---:|:---:|:---:|:---:|:---:|:---:|:---:|
> | Best raw | 6.739 | 64.56 | 66.54 | 26.40 | 3.03 | 32.45 | 24.12 | 36.18 |
> | PMA(N=10) | 6.697 | 64.90 | 66.75 | 26.80 | 3.03 | 32.39 | 24.82 | 36.45 |
> | EM(α=0.4) | 6.643 | 65.17 | 67.41 | **27.40** | **3.97** | 32.67 | 25.70 | 37.05 |
> | EM(α=0.8) | **6.592** | **65.46** | **67.81** | **27.40** | 3.95 | **33.45** | **26.20** | **37.38** |
> | EM(α=1.2) | 6.680 | 64.91 | 66.96 | 26.80 | 2.98 | 32.83 | 25.50 | 36.66 |
>
> **EM(α=0.8) consistently outperforms both the best raw checkpoint and PMA across all six benchmarks.** We emphasize that Extra-Merge is a fully **training-free** method that processes no new tokens; all improvements stem purely from loss landscape geometry. On these six benchmarks, Extra-Merge's average improvement (+1.20) is **over 4× that of PMA** (+0.27), the strongest existing training-free baseline. We hope these extended results address the reviewer's concern on evaluation breadth and task difficulty.
>
> **S3.2: Evaluation on more recent models (OLMo)**
>
> A3.2: We sincerely thank the reviewer for this valuable suggestion. We evaluate Extra-Merge on OLMo-7B using the same configuration as in the paper (N=10, K=4), with the Dolma-v1.5 validation set for α selection. We take `step550000-tokens2433B` as the last checkpoint in the merging window, corresponding to a late-stage checkpoint from the publicly released series. Ten downstream tasks are evaluated; all are zero-shot except GSM8K (8-shot, greedy, strict-match) and MMLU (0-shot, evaluated via lm-eval with batch size 4, bf16).
>
> **Table 2.** OLMo-7B results.
>
> | Model | PPL↓ | ARC-C | ARC-E | PIQA | Hella. | Wino. | BoolQ | OBQA | TQA | GSM8K | MMLU | Avg. |
> |:---|:---:|:---:|:---:|:---:|:---:|:---:|:---:|:---:|:---:|:---:|:---:|:---:|
> | Best raw | 7.443 | 40.27 | 67.94 | 79.38 | 75.00 | 66.32 | 71.57 | 42.40 | 36.80 | 7.13 | 27.12 | 51.39 |
> | PMA | 7.403 | 40.79 | 68.00 | 79.58 | 74.81 | 66.76 | 71.89 | 42.60 | 36.70 | 6.60 | 27.41 | 51.51 |
> | EM(α=1.0) | **7.327** | **41.19** | **68.97** | **80.20** | **75.46** | **67.46** | **72.27** | **43.00** | 36.68 | **7.88** | **28.92** | **52.20** |
>
> **Extra-Merge achieves the lowest Dolma validation PPL and the best downstream score on 9 out of 10 tasks.** Its average improvement (+0.81) over the best raw checkpoint is approximately **7× that of PMA** (+0.12), further confirming Extra-Merge's effectiveness and robustness on modern architectures. We sincerely thank the reviewer for this suggestion and will incorporate these results into the revised manuscript. We hope these results adequately address the reviewer's concern.
>
> **S4: Effect of the number of merged checkpoints**
>
> We sincerely thank the reviewer for this insightful question. Fig. 4 in the paper demonstrates that the rank-1 linearity of the checkpoint subspace increases rapidly with N and already reaches a high level at N≈6. Since this linearity is precisely the geometric property Extra-Merge exploits, we expect the method to become effective around the same threshold. To verify this, we ablate N on our GPT-2 Small experiment (step 20k):
>
> **Table 3.** GPT-2 Small validation loss under varying N.
>
> | | Raw (step 20k) | EM (N=4) | EM (N=6) | EM (N=8) | EM (N=10) |
> |:---|:---:|:---:|:---:|:---:|:---:|
> | Val Loss | 3.1940 | 3.1902 | 3.1862 | 3.1853 | 3.1847 |
>
> **Extra-Merge already yields a clear improvement at N≥6, and the gain saturates thereafter**, fully consistent with the linearity trend in Fig. 4. We sincerely thank the reviewer for this insightful question and will include this ablation in the revised manuscript. We hope this experiment sufficiently addresses the reviewer's concern.

---

> > ### Author Rebuttal · Reviewer_zoPw · 2026-04-02
> >
> > Thanks for the reply. I'd like to keep my positive attitude to this paper because I think the authors addressed my concerns.

---

### Official Review · Reviewer_mCjZ · 2026-03-18

**Soundness:** 3
**Presentation:** 3
**Significance:** 2
**Originality:** 3
**Overall Recommendation:** 4
**Confidence:** 3

**Summary:**

This paper investigates why checkpoint averaging improves LLM pre-training and whether the resulting geometric structure can be further exploited. The authors show that merged checkpoints collapse onto an approximately rank-1 subspace (PC1 > 94% variance) with nearly monotonic loss descent, in contrast to the oscillatory raw trajectory. Based on this, they propose Extra-Merge, which extrapolates along the PCA direction via a 1D line search to reduce loss without gradient updates. Theoretically, this is grounded in a river-valley landscape analysis where averaging acts as a geometric low-pass filter. Experiments on GPT-2, LLaMA (124M–2B), and Pythia-12B show consistent improvements over PMA baselines, with gains also transferring to the Muon optimizer.

**Compliance With Llm Reviewing Policy:**

Affirmed.

**Key Questions For Authors:**

How is α selected for the Pythia-12B downstream experiments? Is it chosen using validation loss, a separate development benchmark set, or the same four-task average reported in the table? This point is critical for judging the credibility of the downstream results.

**Limitations:**

The authors acknowledge compute costs but do not discuss methodological limitations such as the strong theoretical assumptions, narrow downstream evaluation, or the overhead of the line search procedure.

**Strengths And Weaknesses:**

Strengths
1. , the paper builds a full chain from empirical diagnosis to geometric interpretation, theoretical explanation, algorithm design, and experimental validation. This makes the contribution intellectually cleaner and more compelling than a purely empirical trick.
2. The contrast between the U-shaped interpolation for raw checkpoints and the nearly monotonic interpolation for merged checkpoints is very intuitive.

Weaknesses
1. The most serious issue is ambiguity in the downstream model-selection protocol, which raises possible evaluation leakage concerns. In Section 6.3, the paper states that Pythia-12B results are obtained “using average score as the indicator,” while the appendix describes selecting the extrapolation coefficient α via a validation-based 1D search over discrete candidates. If the reported four-task average itself is used to choose α, that would amount to tuning on the test benchmarks and would significantly weaken the downstream claims. Even if this is not what the authors intended, the current presentation is ambiguous and needs to be clarified.
2. The downstream gains are real but modest, and the evaluation is somewhat narrow. The Pythia-12B results cover only four zero-shot tasks, with an average improvement of +0.59 over raw and +0.40 over best PMA. The larger gains seem concentrated on ARC-Challenge and ARC-Easy, while HellaSwag and PIQA improve only slightly. This makes it difficult to assess how robust or broadly useful the method is on downstream tasks.
3. Although the method does not require extra gradient updates, it still requires repeated forward evaluations along the extrapolation direction. The appendix discusses the PCA cost, but the paper does not clearly quantify the full wall-clock or validation-time overhead introduced by the search procedure, which matters for practical adoption.

---

> ### Author Rebuttal · Authors · 2026-03-28
>
> We sincerely thank the reviewer for recognizing our work as building a **"full chain from empirical diagnosis to geometric interpretation, theoretical explanation, algorithm design, and experimental validation."** We address each concern below.
>
> **W1 & KQ1: Ambiguity in the downstream model-selection protocol**
>
> A1: We sincerely appreciate the reviewer for raising this important point. We clarify that α is selected **exclusively using pretraining validation loss**, with no access to downstream scores. The phrase "using average score as the indicator" in Section 6.3 means we use average downstream score to **demonstrate Extra-Merge's effectiveness**; it plays no role in α selection. We apologize for the ambiguity.
>
> Extra-Merge operates within the pretraining distribution, seeking a model with lower val loss and a flatter landscape position (cf. Fig 5), independent of any downstream benchmark. In all experiments, we **always** select α by validation loss on the pretraining corpus, for Pythia-12B, the Pile validation split. Table 1 **in A2** jointly reports Pile PPL and downstream metrics, showing that α=0.8 (selected solely by lowest PPL) also achieves the best downstream performance. We sincerely thank the reviewer for this reminder; we will state explicitly in Sec. 6.3 that α is chosen by minimizing Pile validation perplexity. We hope this clarification fully addresses the concern.
>
> **W2: Evaluation scope and robustness**
>
> A2: We sincerely thank the reviewer for this valuable suggestion. We first note that Extra-Merge is fully **training-free**, processing no new tokens; all gains stem purely from loss landscape geometry, demonstrating meaningful value as a zero-cost technique.
>
> Following your valuable suggestion, we additionally evaluate Pythia-12B on six additional benchmarks: Winogrande, BoolQ, OBQA, GSM8K, TruthfulQA-MC2, and MMLU, using lm-eval (v0.4.10.dev0, bf16). All tasks are 0-shot except GSM8K (5-shot, greedy). Metrics: accuracy for Winogrande/BoolQ/MMLU, acc\_norm for OBQA, exact\_match for GSM8K, MC2 for TruthfulQA. EM uses default settings (K=4, N=10). Pile validation PPL is included to show our α-selection criterion.
>
> **Table 1.** Extended Pythia-12B results. EM = Extra-Merge.
>
> | Model | Val_PPL↓ | Wino. | BoolQ | OBQA | GSM8K | TQA | MMLU | Avg. |
> |:---|:---:|:---:|:---:|:---:|:---:|:---:|:---:|:---:|
> | Best raw | 6.739 | 64.56 | 66.54 | 26.40 | 3.03 | 32.45 | 24.12 | 36.18 |
> | PMA(N=10) | 6.697 | 64.90 | 66.75 | 26.80 | 3.03 | 32.39 | 24.82 | 36.45 |
> | EM(α=0.4) | 6.643 | 65.17 | 67.41 | **27.40** | **3.97** | 32.67 | 25.70 | 37.05 |
> | EM(α=0.8) | **6.592** | **65.46** | **67.81** | **27.40** | 3.95 | **33.45** | **26.20** | **37.38** |
> | EM(α=1.2) | 6.680 | 64.91 | 66.96 | 26.80 | 2.98 | 32.83 | 25.50 | 36.66 |
>
> **Key results: (1)** α=0.8, selected solely by lowest Pile Val PPL, also achieves the best downstream performance, confirming that validation-guided α selection is effective and sufficient. **(2)** EM(α=0.8) **consistently outperforms** both the best raw checkpoint and PMA across all six additional benchmarks. We hope these extended results alleviate the reviewer's concern on evaluation breadth and robustness.
>
> To further demonstrate Extra-Merge's effectiveness modern architectures, we present OLMo-7B results with consistent downstream gains in our response to **Reviewer zoPw (Table 2)**; we refer the reviewer there due to space constraints.
>
> **W3: Wall-clock overhead**
>
> We sincerely thank the reviewer for this valuable suggestion. Extra-Merge's post-processing is **lightweight and fast**. Appendix B.5 detailed the direction computation but did not make end-to-end overhead explicit. We clarify below.
>
> We use the **Gram-matrix trick**: for K centered checkpoints (K=4), we compute a K×K Gram matrix via **shard-wise streaming** and eigen-decompose this small matrix. Measured costs on Pythia-12B:
>
> | Stage | Operation | Peak Memory | Measured (12B) |
> |:---:|:---|:---|:---|
> | S1 | PMA (averaging) | $\mathcal{O}(d)$ | Pre-computed to disk |
> | S2 | Gram matrix (streaming) | $\mathcal{O}(K \cdot \vert \text{shard} \vert)$ | 151.6 s, 33.5 GB$^\dagger$ |
> | S3 | Eigen-decompose $G \in \mathbb{R}^{K \times K}$ | $\mathcal{O}(K^2)$ | $\approx 0$ s |
> | S4 | Recover $\hat{u}_1$ (streaming) | $\mathcal{O}(d + \vert \text{shard} \vert)$ | 82.8 s, 33.5 GB$^\dagger$ |
> | S5 | 1D search + val loss | $\mathcal{O}(d)$ | Same as normal eval |
>
> $^\dagger$ Peak **CPU RAM**; reducible via finer-grained streaming.
>
> Key points: (1) Peak RAM scales **linearly with model size** (33.5 GB at 12B; single-digit GB for ≤2B). (2) **S2–S4 run only once**; afterwards Extra-Merge is scalar extrapolation along \(\hat{u}_1\) plus standard validation passes. Total CPU post-processing: **~4 min for Pythia-12B**, negligible vs. pre-training cost. We sincerely thank the reviewer for raising this practical concern and will add this detail to revised Appendix B.5. We hope this breakdown fully addresses the concern.

---

> > ### Author Rebuttal · Reviewer_mCjZ · 2026-04-04
> >
> > Thank you for the detailed rebuttal. I appreciate the clarification on 𝛼, as well as the additional downstream results and the discussion of the post-processing overhead.
> >
> > Overall, my main concerns have been adequately addressed. These clarifications strengthen my confidence in the paper, but they do not materially change my overall assessment of its significance, so I keep my original positive score.

---

### Decision · Program_Chairs · 2026-04-30

**Decision:**

Accept (regular)

**Comment:**

This paper identifies the phenomenon that training trajectories in the late stage almost live in a rank-1 subspace. Based on this observation, the paper proposes Extra-Merge, a model merging strategy that extrapolates along this subspace to minimize loss. Overall, the reviewers view this as a solid paper, with interesting findings, a plausible theoretical explanation, and thorough experimental validation. While one reviewer noted that the theoretical foundation is not sufficiently strong, the main contribution of the paper lies in identifying the rank-1 collapse phenomenon and developing an improved model merging method based on this observation. We believe these strengths outweigh the limitations and recommend acceptance.